# Speculative Knowledge Distillation: Bridging the Teacher-Student Gap Through Interleaved Sampling

**Wenda Xu**[1][*]  **Rujun Han**[2]  **Zifeng Wang**[2]  **Long T. Le**[2]  **Dhruv Madeka**[2]  **Lei Li**[4]
**William Yang Wang**[1]  **Rishabh Agarwal**[3]  **Chen-Yu Lee**[2]  **Tomas Pfister**[2]
[1]UC Santa Barbara  [2]Google Cloud AI Research  [3]Google DeepMind  [4]CMU

## Abstract

Recent advances in knowledge distillation (KD) have enabled smaller student models to approach the performance of larger teacher models. However, popular methods such as supervised KD and on-policy KD, are adversely impacted by the knowledge gaps between teacher-student in practical scenarios. Supervised KD suffers from a distribution mismatch between training with a static dataset and inference over final student-generated outputs. Conversely, on-policy KD, which uses student-generated samples for training, can suffer from low-quality training examples with which teacher models are not familiar, resulting in inaccurate teacher feedback. To address these limitations, we introduce Speculative Knowledge Distillation (SKD), a novel approach that leverages cooperation between student and teacher models to generate high-quality training data on-the-fly while aligning with the student's inference-time distribution. In SKD, the student proposes tokens, and the teacher replaces poorly ranked ones based on its own distribution, transferring high-quality knowledge adaptively. We evaluate SKD on various text generation tasks, including translation, summarization, math, and instruction following, and show that SKD consistently outperforms existing KD methods across different domains, data sizes, and model initialization strategies. The code and data are released at https://github.com/google-research/google-research/tree/master/speculative_kd.

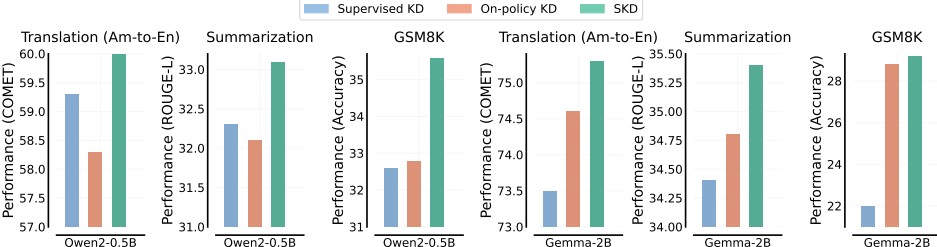

**Figure 1:** SKD outperforms supervised and on-policy KD for our tested tasks: Assamese-to-English translation, dialogue summarization, and arithmetic reasoning. For teacher models, we employ supervised FT Gemma-7B-it and Qwen-7B-it as teacher models (it means instruction-tuned); for student models, we use Gemma-2B and Qwen-0.5B (either instruction-tuned or supervised FT student models checkpoints depending on the best performance). Supervised KD is trained on ground-truth outputs, while on-policy KD uses self-generated data. All models use greedy decoding for evaluation.

## 1 Introduction

Text generation capabilities of large language models (LLMs) have seen continuous improvement, largely driven by scaling up the number of parameters and the amount of training data (Kaplan

[*]Work done as a student researcher at Google Cloud AI Research. Correspondence to: Wenda Xu (wendaxu@cs.ucsb.edu), Rujun Han (rujunh@google.com), Rishabh Agarwal (rishabhagarwal@google.com), Chen-Yu Lee (chenyulee@google.com)

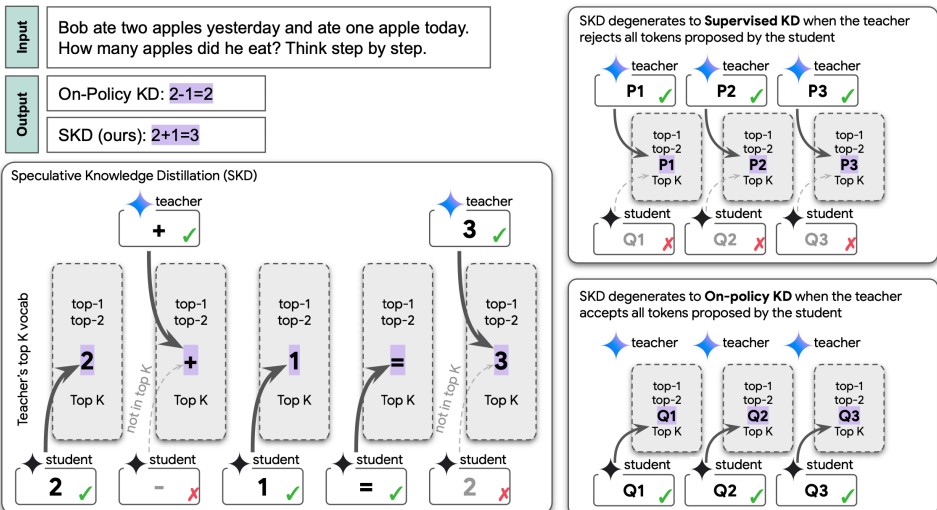

**Figure 2: Overview of Speculative Knowledge Distillation (SKD)** on an arithmetic reasoning task. **Left:** SKD addresses the limitations of on-policy knowledge distillation (KD) by filtering out low-quality student samples and replacing them with teacher generated tokens. **Right:** illustration of how SKD generalizes to both supervised KD (replacing with all teacher tokens) and on-policy (accepting all student tokens).

et al., 2020; Anil et al., 2023; Dubey et al., 2024; OpenAI et al., 2024). However, the substantial inference-time costs and memory footprint associated with LLMs present significant challenges for practical deployment (Agarwal et al., 2024). Therefore, compressing LLMs while maintaining their performance is crucial for real-time practical applications.

Knowledge Distillation (KD) (Hinton et al., 2015) is a widely used method to compress LLMs by transferring knowledge from a larger teacher model to a smaller student model. Traditional KD approaches, such as supervised KD (Sanh et al., 2020) and SeqKD (Kim & Rush, 2016b), rely on a static dataset of outputs to train the student model. However, this fixed dataset can lead to a distribution mismatch between the training data and the student's generated samples at inference time, hindering the student's learning. On-policy KD (Agarwal et al., 2024) attempts to address this train-inference mismatch by training the student on its self-generated samples, with the teacher provides feedback through its token-level probabilities. However, the student may generate low-quality samples that are out-of-distribution (OOD) for the teacher, especially early in the training, leading to inaccurate teacher feedback. Additionally, the performance of on-policy is highly sensitive to the student's initialization, as illustrated in Figure 4.

In this work, we introduce **Speculative Knowledge Distillation** (SKD) that uses interleaved sampling between teacher and student to overcome the challenges faced by supervised and on-policy distillation (Figure 2). Similar to on-policy KD, SKD utilizes student-generated samples to address the train-inference mismatch. However, to mitigate the issue of low-quality student samples, SKD filters out intermediate tokens that the teacher is unlikely to produce and instead re-samples them from the teacher, drawing inspiration from speculative decoding (Leviathan et al., 2023). The design of SKD is supported by theoretical results in imitation learning (Ross & Bagnell, 2010; Ross et al., 2011), which suggest that gradually shifting the sample generation from the teacher to the student is an effective approach. Early in training, when the student model is likely to propose many out-of-distribution tokens, SKD functions similarly to supervised KD, replacing low-quality tokens with those from the teacher. As training progresses and the student's sample quality improves, SKD behaves more like on-policy KD, accepting many of the student's proposed tokens.

SKD significantly outperforms supervised and on-policy distillation for LLMs, as shown in Figure 1. Specifically, when distilling Gemma-7B to Gemma-2B, SKD achieves substantial gains compared to supervised fine-tuning in low-resource machine translation (41.8%), summarization (230%), and arithmetic reasoning (160%). Furthermore, training SKD on instruction-following data leads to impressive results on held-out MATH (Hendrycks et al., 2021) (198%) and $GSM_{plus}$ (Li et al., 2024) (360%) testing suites. To provide a comprehensive understanding of SKD's capabilities, we conduct a systematic evaluation under varying conditions and offer practical guidelines for tokens acceptance

criteria from the teacher model, considering factors such as task type and model initialization. Our key contributions are:

- **A novel KD approach** that utilizes on-policy student samples *likely* to be generated by the teacher, mitigating the issues of low-quality samples in on-policy KD and dynamically switching between supervised and on-policy KD.

- **Superior performance** across various text generation tasks, including summarization, translation, and reasoning tasks, in both task-specific and task-agnostic scenarios.

- **Robustness and adaptability** across different model initialization, data sizes, and model families.

## 2 PRELIMINARIES

**Language Models.** An auto-regressive language model, denoted as $M$, predicts the probability distribution of the next token in a sequence. Conditioned on an input sequence $X$ and the generated prefix $y_{<i} = \{y_1, y_2, ..., y_{i-1}\}$ consisting of $(i-1)$ tokens from a vocabulary $V$, $M$ calculates the probability of the $i$-th token, $p_{y_i} = M(y_i|X, y_{<i})$. The auto-regressive language model M computes a logit $l$ at the $i$-th token position. To obtain the probability $p_{y_i}$, we apply the softmax function with temperature $t$: $p_{y_i} = \frac{exp(l_i/t)}{\sum_{j=1}^{|V|} exp(l_i/t)}$. The temperature $t$ introduces randomness into the text generation process. A higher temperature leads to more diverse sample output, while a lower temperature results in more focused and deterministic generation. In practical implementations, the vocabulary $V$ is often truncated using top-k sampling (Radford et al., 2019), which improves text generation compared to searching over the entire vocabulary distribution. We draw inspiration from top-k sampling to develop our acceptance criteria for student-generated tokens. We include background of divergence metrics in the Appendix A.

**Speculative Decoding (Leviathan et al., 2022; Chen et al., 2023)** accelerates the decoding process by employing a smaller draft model to generate token candidates, which are then verified by larger approximation models. Our interleaved sampling is inspired by speculative decoding but introduces key differences. While speculative decoding ensures that final sampled tokens adhere to the larger model's distribution, our method directly assesses the feasibility of student-proposed tokens within the teacher's top $K$ tokens (See Sec 3.2).

## 3 SPECULATIVE KNOWLEDGE DISTILLATION FOR LANGUAGE MODELS

**Problem Setup.** Given a teacher LM $M_t$ and a student LM $M_s$, with differing model capacities, our goal is to train $M_s$ to mimic the behavior of $M_t$ on a specific task $T$. We assume that $M_s$ has learnable parameters $\theta_s$, while $M_t$'s parameters $\theta_t$ are fixed. For task $T$, we have a set of prompts $\{x\}$ and corresponding output sequences $\{y\}$, which can be generated by either model or provided as ground truth. To measure the divergence between the token-level distributions of $M_t$ and $M_s$ for a given input-output pair $(x, y)$, we define the following metric:

$$D(M_t||M_s)(y|x) = \frac{1}{L_y} \sum_{i=1}^{L_y} D(M_T(.|y_{<i}, x)||M_s(.|y_{<i}, x)) \tag{1}$$

where $L_y$ is the length of the output sequence $y$, and $i$ indicates decoding steps. Our training objective is to minimize $D(M_t||M_s)(y|x)$ such that $M_s$ can effectively imitate $M_t$ on task $T$.

### 3.1 BASELINE KNOWLEDGE DISTILLATION APPROACHES

**Supervised FT** is a training method where a student policy is trained on a fixed dataset of input-output pairs $(x, y)$. The objective is to minimize the negative log-likelihood of the student's predictions: $L_{SFT} = -\log p_s(y|x)$. It's also a common initialization strategy for more advanced knowledge distillation techniques like Supervised KD, On-Policy KD, and SKD.

**SeqKD (Kim & Rush, 2016a)** is a supervised fine-tuning method that trains on sequences generated by a teacher model using maximum likelihood.

**Supervised KD (Sanh et al., 2020)** trains a student model to mimic the token-level probability distribution of a teacher model over a fixed ground truth dataset. This is achieved by minimizing the distance between the student's and teacher's predictions, as defined in Equation 1. The ground truth labels, denoted by y, are obtained from the fixed dataset.

---

**Algorithm 1** Speculative knowledge distillation

---

**Require:** Student LLM $M_s$, Teacher LLM $M_t$, Prompt dataset $\{x_j\}_{j=1}^N$, Decoding length $\alpha$.
1: **Hyperparameters**: Top $K$ for token acceptance, Divergence metric $D$,
2: **for** $step := 1$ to $N$ **do** ▷ We assume batch size $:= 1$ to simplify the illustration
3:   **for** $i := 1$ to $\alpha$ **do**
4:     $y_i \sim M_s(.|y_{<i}, x_j)$ ▷ Sample $y_i$ from student $M_s$
5:     **if** $y_i \not\subset top_K(M_t(.|y_{<i}, x_j))$ **then** ▷ If $y_i$ is not within top $K$ token of teacher $M_t$
6:       $y_i \sim M_t(.|y_{<i}, x_j)$ ▷ We replace student token to teacher's resampled token
7:     **end if**
8:   **end for**
9:   Apply gradient descent to minimize the minibatch loss $D(M_t||M_s)(y|x)$ in equation 1
10: **end for**

---

**On-Policy KD (Agarwal et al., 2024)** addresses the training-inference mismatch in student models by sampling target tokens directly from the student's own output. This approach calculates both the teacher and student's token probabilities and employs Equation 1 to train the student model on its self-generated data.

**ImitKD (Lin et al., 2020)** is a hybrid approach that can be viewed as naively combining supervised and on-policy KD. During training, it randomly selects samples from either the ground truth dataset or the student model's own output with equal probability. This mixed dataset is then used to train the student model, optimizing the distance metric defined in Equation 1.

## 3.2 SPECULATIVE KNOWLEDGE DISTILLATION

Speculative knowledge distillation (SKD) draws two important inspirations from imitation learning (Ross et al., 2011; Ross & Bagnell, 2010): 1) Directly taking samples from a fixed dataset or sampling from teacher can cause the state distribution to be unrealistically good. As a result, student model never learns to correct previous mistakes, leading to poor performance at test time. 2) Directly drawing samples from student can result in low quality samples that can be out-of-distribution for the teacher, resulting in inaccurate feedback when assigning logits to the student. SKD addresses above limitations by using major components from speculative sampling (Leviathan et al., 2023): 1) student model proposes on-policy sample candidate, and 2) the teacher model evaluates these proposed samples concurrently and accepts only those that it deems likely to generate. After speculative sampling, SKD calculates teacher and student's token probabilities and employ equation 1 to train the student model (L9 in Algorithm 1).

**Interleaved On-Policy Sampling** Given a prompt $x$, student model $M_s$ will generate a sequence of tokens, $(y_i, ..., y_{i+\gamma})$, following its own distribution $M_s(.|y_{<i}, x)$ (L4 in Algorithm 1). The teacher model $M_t$ will evaluate this sequence, accepting or rejecting each token based on pre-defined criteria, which resembles speculative decoding (Leviathan et al., 2023). If a token $y_i$ is rejected, the subsequent tokens $(y_{i+1}, ...)$ will be discarded, and $y_i$ will be resampled from $M_t$'s distribution $M_t(.|y_{<i}, x)$ (L5-7 in Algorithm 1). The student model will then continue to generate tokens on-policy based on $M_s(.|y_{<i}, x, y_i)$ for the remaining $\gamma$ tokens. This sample generation process is particularly advantageous for auto-regressive models, where early errors can propagate and affect future predictions (Ross & Bagnell, 2010). By focusing on on-policy samples that are likely to be generated by the teacher model, we can effectively utilize limited student capacity to mimic teacher.

**Token Acceptance Criteria** Using the default acceptance criteria in speculative decoding corresponds to sampling from teacher's distribution, which would degenerate to supervised KD. Instead, we are interested in generating student samples that are likely under the teacher. To do so, we draw inspiration from top-k sampling (Radford et al., 2019) that draws samples using only the k highest probability tokens from the entire vocabulary $V$ at each generation step. Following this intuition, we define our acceptance criteria based whether the student token $y_i$ falls into the top $K$ tokens of teacher's distribution $M_t(.|y_{<i}, x)$. Similar to speculative decoding, the teacher $M_t$ can evaluate sequence $(y_i, ..., y_{i+\gamma})$ proposed by student in parallel and return to student with the token position that discarded. We replace the discarded student token by re-sampling a token from the teacher's distribution. Note that in Algorithm 1, we simplify the SKD presentation by setting $\gamma = 1$, but in practice use $\gamma = 5$ for efficient implementation.

**Sample Transition During SKD Training.** In the early stages of training, student models often propose low-quality samples that are frequently rejected by the teacher. This initial behavior resembles supervised KD, where the student is explicitly corrected and guided towards the teacher's generated samples. As the student's training progresses, the quality of its proposed samples improves, leading to a more on-policy interaction with the teacher. SKD dynamically adjusts the balance between supervised and on-policy KD based on the distribution gap between the teacher and student models. This adaptive approach allows the student to efficiently learn from the teacher's samples. Moreover, SKD is a flexible framework that can degenerate both supervised and on-policy KD as special cases. In Figure 2, when the student's samples are consistently rejected or accepted by the teacher, SKD effectively degenerates to supervised or on-policy KD, respectively. This demonstrates the generality and versatility of SKD as a unified approach to knowledge distillation.

## 4 EXPERIMENTAL SETUP

**Student and Teacher Models**. We conducted experiments using two model families, GEMMA1 (Team et al., 2024) and QWEN2 (Yang et al., 2024a). We use supervised fine-tuned (SFT) GEMMA-7B-IT (IT indicates instruction-tuned) and QWEN-7B-IT as the teacher models, while GEMMA-2B-IT and QWEN-0.5B-IT were employed as student models. We further leveraged SFTed student models to compare KDs under different model initializations.

**Hyperparameters**. For all fine-tuning process, we use learning rate 1e-5, warmup ratio 0.1 and 0.1 drop out rate for all fine-tuning processes. All SFT checkpoint is trained for three epoches and we select the checkpoint with the lowest validation loss. Details about SFT training dataset can be found in Section 4.1. Without parameter search, we fixed acceptance criteria $K$ to be 25 throughout main experiment and ablation ($K$=25 is a common choice for top-k sampling (Radford et al., 2019). See Appendix B for detailed study). We include training details and our hyper-parameter choice of baselines and SKD in the Appendix Section D.

**Baselines.** We conduct SKD over widely accepted KD approaches listed in Section 3.1: Supervised Fine-tuning, Supervised KD, on-policy KD and ImitKD. We compared SeqKD to supervised fine-tuning (SFT) using ground truth data and found that SFT consistently outperformed SeqKD. Therefore, we did not include SeqKD results.

### 4.1 DATASET AND EVALUATION.

We evaluated SKD and baseline methods across three generation tasks: low-resource translation, dialogue summarization, and arithmetic reasoning. Task-specific KD often operates with training data around 1K samples (Quteineh et al., 2022). Thus, we randomly draw 1K samples from our task data, and further test low-data regime with 100 samples. Additionally, we performed experiments using 1K and 10K on a task-agnostic instruction-following task in the math domain. It's important to note that different baseline KD methods have access to different training data. Supervised KD, SFT, and ImitKD have access to both $(x, y)$ pairs from the ground truth dataset, while on-policy and SKD only see the prompt $x$ and self-generate the target label during training. Since our assumption that the teacher model should be an expert in a specific task, we used full training data for each task to supervise the FT teacher model. For SFTed student initialization, we supervise the FT student model with our constructed training set $x, y$ (around 1000 instances). For reproducibility, all baselines and SKD use greedy decoding for evaluation. We include all results of supervised FT teacher model, supervised FT student model and instruction tuned student models in the Appendix E.

**Low Resource Translation.** We utilize the Flores-200 (Team, 2022) Assamese-to-English translation dataset for low-resource translation. Due to the scarcity of Assamese-to-English training data, we employ the Flores-200 development set (997 instances) as our training set. Additionally, we split the Flores-200 testing set (1012 instances) into a development set (500 instances) and a testing set (512 instances) for evaluation. We use SOTA learned metric COMET (Rei et al., 2022) to evaluate the translation quality. We use 997 instances from our constructed training set to SFTed both teacher and student models.

**Dialogue Summarization.** For dialogue summarization, we utilize the DialogSum dataset (Chen et al., 2021). We randomly sample 1K instances from the DialogSum training set to create our input-output pairs $(x, y)$. For evaluation, we employ the dataset's development set (500 instances)

and test set (1500 instances). To assess summary quality, we use the ROUGE-L metric (Lin, 2004). We Supervised FT our teacher models on 10K instances from the DialogSum training set, and our student models on the 1K instances from our constructed training set.

**Arithmetic Reasoning.** For arithmetic reasoning, we utilize the GSM8K dataset (Cobbe et al., 2021). We split the GSM8K training set into a development set (473 instances) and a training set (7K instances). We randomly sample 1K instances from the training set to create our input-output pairs $(x, y)$. For evaluation, we employ the development set and the GSM8K test set (1319 instances). We assess the quality of the generated output using answer accuracy. We Supervised FT our teacher models on 7K instances from the GSM8K training set, and our student models on the 1K instances from our constructed training set.

**Math Instruction Following.** For math instruction following, we utilize the UltraInteract dataset (Yuan et al., 2024). We randomly sample 11K instances from the UltraInteract training set, dividing them into a training set (10K instances) and a development set (1K instances). For evaluation, we employ the GSM$_{plus}$ (Li et al., 2024), Math (Hendrycks et al., 2021), Asdiv (Miao et al., 2020) and SVAMP (Patel et al., 2021) test sets as held-out sets. We Supervised FT our teacher models on 10K instances from the UltraInteract training set, and our student models on the 1K instances from our constructed training set. We also tested the setting with 10K training instances for student model.

We explored two types of knowledge distillation: task-specific (Quteineh et al., 2022) and task-agnostic (Agarwal et al., 2024). In task-specific distillation, training and inference data share the same format, and the task is known during training. In task-agnostic distillation, only the domain is known, while the specific task varies during deployment. We categorized distillation of translation, dialogue summarization, and arithmetic reasoning as task-specific and distillation of math instruction following as task-agnostic. For both math instruction following and arithmetic reasoning, we use chain-of-thought (Wei et al., 2023) prompts to generate answer. All prompt formats can be found in Appendix F. Details about parsing and evaluation for all tasks can be found in the Appendix G.

## 5 RESULTS

We validate SKD's effectiveness through comprehensive studies, including task-specific (Section 5.1) and task-agnostic distillation (Section 5.2), different model initializations (Section 5.3), low-data regimes (Section 5.4), and an ablation study on two-staged training (supervised KD followed by on-policy KD) (Section 5.5).

**Table 1:** SKD outperforms baseline KDs at in-domain tasks: low resource Assamese-to-English translation, dialogue summarization, arithmetic reasoning. We use GEMMA-2B and QWEN-0.5B instruction tuned model (w/o SFT) as students and SFTed GEMMA-7B-IT and QWEN2-7B-IT as teachers. $K$=25 for SKD. We conducted a permutation significance test for SKD against all baselines for translation and summarization. SKD outperforms all baselines (p<0.05) significantly except for SKD vs ImitKD at Gemma-7B to Gemma-2B-IT on summarization and SKD vs supervised KD/SFT at Qwen-7B to Qwen-0.5B-it on translation.

| | Supervised FT GEMMA-7B to GEMMA-2B | | | Supervised FT QWEN2-7B to QWEN2-0.5B | | |
|---|---|---|---|---|---|---|
| | Translation | Summarization | GSM8K | Translation | Summarization | GSM8K |
| Baselines | COMET | ROUGE-L | Acc | COMET | ROUGE-L | Acc |
| SFT | 72.5 | 31.7 | 18.7 | **57.6** | 29.2 | 31.8 |
| Supervised KD | 73.3 | 34.1 | 22.5 | 57.5 | 31.3 | 35.0 |
| On-policy KD | 36.1 | 34.1 | 25.3 | 53.0 | 31.2 | 33.9 |
| ImitKD | 74.8 | 34.9 | 26.2 | 55.9 | 30.9 | 33.2 |
| SKD | **75.3** | **35.0** | **29.1** | 57.1 | **31.7** | **36.6** |

### 5.1 TASK-SPECIFIC DISTILLATION

In this section, we investigated task-specific distillation, where the training and inference prompt-response sets share the same format, and the task is known during training. We study two model families, supervised FT GEMMA-7B to instruction tuned GEMMA-2B (GEMMA-2B-IT) and supervised FT QWEN2-7B to instruction tuned QWEN2-0.5B (QWEN2-0.5B-IT). As shown in Table 1, the performance of supervised KD and on-policy KD is highly task-dependent, likely due to the model's initial familiarity with each task (see Sec 5.3 for details). SKD, trained on GEMMA-2B-IT, consistently outperformed all baseline KD approaches across three in-domain text generation tasks.

When initialized with QWEN2-0.5B-IT, SKD outperformed all baselines on summarization and GSM8k. We found that on-policy KD consistently underperformed supervised KD and SKD for translation and summarization. This is primarily attributed to the low-quality samples generated by the student model during training. Although ImitKD (partially on-policy) can achieve competitive performance with a stronger student like GEMMA-2B due to its mixed data strategy, it falls behind SKD when using the QWEN-0.5B student due to the lower quality of on-policy samples (See example in Sec 5.3). Our results demonstrate SKD's superiority in five out of six test settings across two model families, highlighting its optimized performance for task-specific distillation.

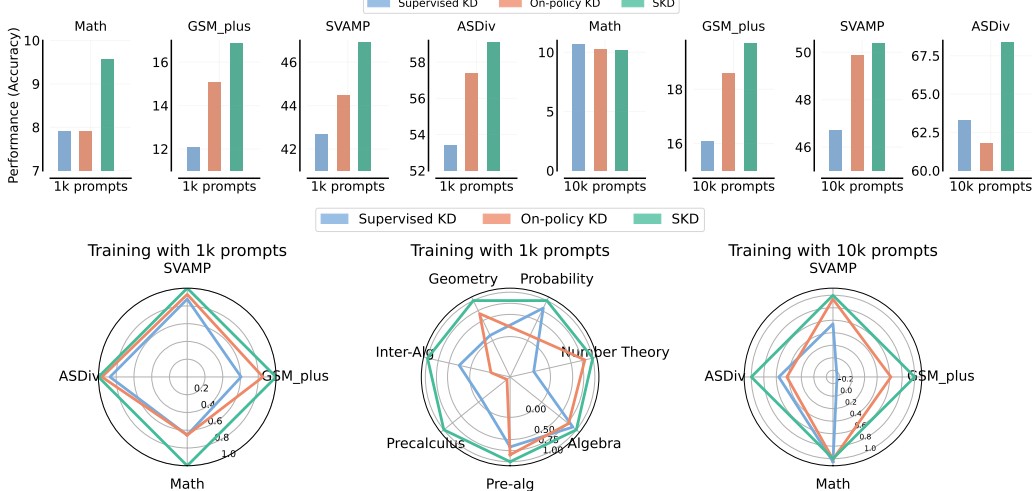

**Figure 3:** SKD outperforms baseline KDs at four held-out testing sets (Math, $GSM_{plus}$, SVAMP and ASDiv). We consider two data size setting, 1k and 10k respectively and report testing accuracy in the first row. In the second row, we calculate the performance gain of SKD over SFT and report the relative performance gains of baselines compared to SKD. As SKD outperforms all baselines in most cases, these values typically range from 0 to 1. In some instances, supervised KD may be outperformed by SFT, resulting in negative values. In the middle figure at the second row, we show that SKD outperforms all baselines under 7 math concepts at math testing set. SKD is performed under $K$=25.

## 5.2 TASK-AGNOSTIC DISTILLATION

To explore task-agnostic instruction following, where the exact nature of the task is unknown during training and can vary during deployment, we trained and evaluated SKD and baselines on a math domain instruction following task. Our training set included diverse math instruction following data, such as MathQA (Amini et al., 2019), math reasoning (Mishra et al., 2022), and tabular processing (Lu et al., 2023). We tested each KD approach's performance on four held-out testing sets, ranging from simple math word problems (ASDiv and SVAMP) to grade-level math problems ($GSM_{plus}$) and competitive math problems (Math).

As illustrated in Figure 3, SKD initialized with GEMMA-2B-IT consistently outperformed supervised KD and on-policy KD across these four held-out sets when trained on 1k prompts. We also calculated the performance gain of SKD over SFT and reported baseline performance gains relative to SKD. Results showed that under 1k prompts, supervised KD and on-policy could approach close performance on simple tasks like SVAMP and ASDiv but still had a 20-40% performance gap compared to SKD (Second row on the left of Figure 3). The middle figure demonstrates SKD's superiority over baselines on the Math task under 7 different concepts. Since the training set was task-agnostic instruction following data, it could be trained on a larger scale in practical scenarios. On the right side of Figure 3, we show that SKD can outperform or match all baseline methods with 10k prompts. Our results demonstrate SKD's generalization capability under task-agnostic distillation. Detailed table result can also be found at Appendix E.

## 5.3 DIFFERENT MODEL INITIALIZATION

In this section, we explore how different student model initialization impact the effectiveness of various KD approaches. As shown in Figure 4, on-policy KD initialized with an instruction-tuned

student model can surprisingly degrade performance over time. This raises the question of whether a teacher's token-wise probability distribution always provides beneficial guidance. Our findings suggest that if the initial student sample quality is insufficient, the student model may become trapped in a sub-optimal state, unable to effectively learn from the teacher's supervision. On-policy KD's low quality sample can be found in Sec 5.3 and Appendix H.

> ### An example of on-policy's training sample vs SKD (QWEN2-0.5B-IT)
>
> **Prompt**: Translate Assamese sentence into English. Assamese: [Assamese Texts]. English:
>
> **Reference**: It was to last for the next 40 years and would be fought for real, by proxy armies, on battlefields from Africa to Asia, in Afghanistan, Cuba and many other places.
>
> **On-policy (COMET: 36)**: This has been a long war, and has been fought in the past, in the past, in the past, in the past, in the past, in the past, in the past, in the past, in the past, in the past, in the past, in the past, in the past......
>
> **SKD (COMET: 72)**: This has been going on for 40 years and is still ongoing, with the possibility of war, a civil war, Afghans in the Balkans, Ethiopia, Kenya, Somalia, and possibly even the United States.

**Table 2:** SKD outperforms all baseline KDs with SFTed student initialization of instruction following model and domain-specific fine-tuned model across three text generation tasks. We use supervised SFTed GEMMA-7B-IT and QWEN2-7B-IT model as teachers and supervised SFT GEMMA-2B-IT and QWEN2-0.5B-IT as students. SKD is performed under $K$=25. We conducted a permutation significance test for SKD against all baselines for translation and summarization. SKD outperforms all baselines (p<0.05) significantly.

| | KD initiates with SFTed Gemma-2B | | | KD initiates with SFTed Qwen-0.5B | | |
|---|---|---|---|---|---|---|
| | Translation | Summarization | GSM8K | Translation | Summarization | GSM8K |
| Baselines | COMET | ROUGE-L | Acc | COMET | ROUGE-L | Acc |
| SFT | 72.5 | 31.7 | 18.7 | 57.6 | 29.2 | 31.8 |
| Supervised KD | 73.5 | 34.4 | 22.0 | 59.3 | 32.3 | 31.9 |
| On-policy KD | 74.6 | 34.8 | 28.8 | 58.3 | 32.1 | 32.2 |
| ImitKD | 74.6 | 34.4 | 26.2 | 59.9 | 32.2 | 31.8 |
| SKD | **75.0** | **35.4** | **29.2** | **60.6** | **33.1** | **33.4** |

As shown in Table 2, SKD with SFT-initialized students consistently outperforms all baselines across both model families. While on-policy KD can narrow the gap with SKD through a two-stage training process (supervised fine-tuning + KD) for GEMMA-2B students, it lags behind SKD for smaller models like QWEN-0.5B. Additionally, in Section 5.4 and Appendix Section I, we demonstrate that SFT can lead to over-fitting in student models under low-data regimes, resulting in sub-optimal performance in post-training KDs. Furthermore, despite some advantages over supervised and on-policy KD with IT-initialized student models, ImitKD does not exhibit clear benefits with higher quality initialized student models. This indicates that SKD's token-level mixing of teacher and student model tokens can achieve more optimal performance than sequence-level mixing.

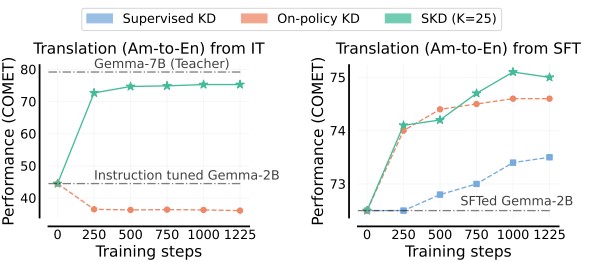

**Figure 4:** Comparison between SKD and baseline KD methods with different model initializations. We find that on-policy methods struggles when the student model starts with a poor initialization, leading to performance degradation and becoming stuck at a low level throughout training. On-policy KD requires student model to have a good initialization. In contrast, SKD outperforms supervised and on-policy KD under both IT and supervised FT Gemma-2B initialization. On-policy KD's low quality sample can be found in Appendix H. SKD is performed under $K$=25.

## 5.4   LOW DATA REGIME

We further investigated SKD under an extremely low data setting with 100 data points per task (10% of the training data). As shown in Table 3, SKD outperformed supervised KD and on-policy KD for all three tasks when initialized with IT and SFTed GEMMA-2B, demonstrating SKD's superior

**Table 3:** SKD outperforms other KD approaches with IT and SFTed student initialization across three text generation tasks at 100 data points. We use supervised SFT GEMMA-7B-IT as teacher and instruction tuned GEMMA-2B and supervised FT GEMMA-2B-IT as students. SKD is performed under $K$=25.

| | KD initiates with IT Gemma-2B | | | KD initiates with SFTed Gemma-2B | | |
| | Translation | Summarization | GSM8K | Translation | Summarization | GSM8K |
| Baselines | COMET | ROUGE-L | Acc | COMET | ROUGE-L | Acc |
|---|---|---|---|---|---|---|
| SFT | 59.8 | 23.0 | 8.11 | 59.8 | 23.0 | 8.11 |
| Supervised KD | 65.8 | 31.7 | 17.0 | 60.9 | 32.0 | 14.4 |
| On-policy KD | 33.7 | 32.2 | 12.1 | 63.4 | 32.4 | 16.1 |
| SKD | **67.0** | **32.3** | **18.8** | **63.8** | **32.6** | **16.7** |

performance in this extreme low-data setting. Notably, both supervised KD and SKD exhibited lower or comparable performance when switching from instruction-tuned to SFTed initialization. This is attributed to unavoidable overfitting during SFT with only 100 data points. In Appendix Section I, we illustrate how overfitting at the SFT stage can lead to suboptimal performance in post-training KDs. This finding suggests that end-to-end SKD, capable of training from any base model and bypassing the SFT stage, offers a more optimal solution than two-stage on-policy KD (SFT+KD) in extreme low-data scenarios.

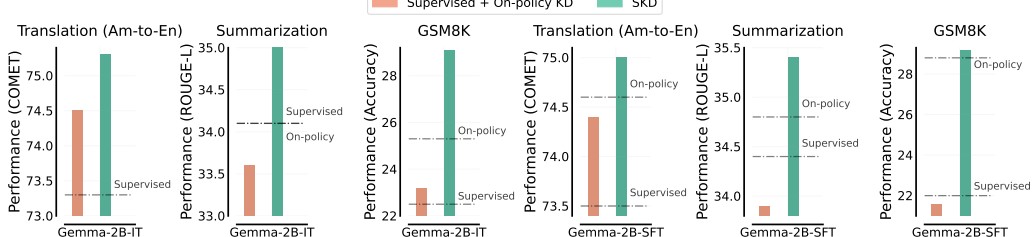

**Figure 5:** Ablation study. We conduct an ablation study by training the student model with supervised KD for the first half and self-generated samples for the second half. Our results demonstrate that this mixed training strategy can mostly be outperformed by either supervised KD or on-policy KD across three tasks and under two model initialization. Moreover, it is significantly outperformed by our SKD. We excluded on-policy KD from the leftmost figure due to its extremely low COMET score.

## 5.5 TWO-STAGED TRAINING

Given that SKD transitions from supervised KD-like behavior at the beginning of training to on-policy KD towards the end, we investigated whether a simple approach of training with supervised KD for the first half of iterations and self-generated samples for the second half could achieve SKD-like performance. We conducted an ablation study by training the student model with supervised KD first, followed by on-policy KD (see Appendix Section J for details). As shown in Figure 5, SKD significantly outperforms this mixed training strategy. Surprisingly, this heuristic baseline was often outperformed by either supervised KD or on-policy KD across three tasks and two model initializations. This is likely due to the data discrepancy encountered during the training process – naively mixing samples from the teacher and student in two different stages lacks natural confidence measurement of the tokens; therefore, simply combining supervised and on-policy KD can even degrade the performance of either approach. This underscores the importance of our adaptive SKD, which implicitly covers intermediate transitions between two model behaviors.

## 5.6 SUMMARY AND ADDITIONAL RESULTS

SKD consistently outperforms baselines in both task-specific and task-agnostic distillation. Unlike on-policy methods, which are sensitive to student-generated sample quality, SKD maintains its advantage even with low-quality student model initialization.

In Appendix B, we demonstrated that a wide range of $K$ values can outperform both supervised and on-policy KD, and that optimal $K$ values can be further explored across tasks. Additionally, Appendix C presents a preliminary experiment on an adaptive $K$ criterion, highlighting a promising direction for future research. Furthermore, as discussed in Appendix I, overfitting in SFT can impede the performance of post-training KD. By adopting an end-to-end approach that bypasses SFT, SKD offers a significant advantage, particularly in low-data scenarios.

Speculative decoding, as described in (Leviathan et al., 2023), is a crucial technique for accelerating LLM inference. The quality of the draft model used in this process directly influences the speed-up achieved. As demonstrated in Appendix K, our SKD-trained draft model significantly outperforms the instruction-tuned base model in terms of token acceptance rate ($71\%$ to $85\%$ higher), leading to a $1.2$X speed-up in speculative decoding.

In Appendices L and M, we analyze the rejection rate of student tokens and estimate the theoretical computational cost of SKD, finding that SKD reduces computation by $50\%$ compared to directly sampling from the teacher model. Additionally, in Appendix N, SKD demonstrates superior validation convergence over baseline methods, achieving the lowest validation loss across tasks and initializations. Finally, in Appendix O, we quantify sample OOD issues across different KD approaches.

## 6 RELATED WORK

**Knowledge Distillation** (KD) (Hinton et al., 2015) is a widely adopted technique to transfer knowledge from a large, complex teacher model to a smaller, more efficient student model. This approach has been successfully applied to autoregressive models (Sanh et al., 2020) and language models in particular (Yang et al., 2024b). There are two primary categories of KD methods: hard label and soft label. Hard label methods train the student model to mimic the teacher's input-output pairs, similar to supervised fine-tuning (Kim & Rush, 2016a; Taori et al., 2023; Chiang et al., 2023; Xu et al., 2023). Various data augmentation techniques, such as fine-tuning on rationales (Hsieh et al., 2023) or instruction-following formats (Wang et al., 2023), have been explored within this framework. Soft label methods, on the other hand, compute logits for both the student and teacher models and minimize a distance metric, such as KL-divergence, between their output distributions (Hinton et al., 2015; Sanh et al., 2020). Samples for computing logits can be generated by either model or both. Our proposed SKD falls under the category of soft label KD. To the best of our knowledge, this work is the first to introduce interleaved teacher-student sampling for soft-label KD.

**Soft-label Knowledge Distillation** can be broadly categorized into supervised KD and on-policy KD based on the source of training data. In supervised KD (Sanh et al., 2020), the training data is fixed, leading to a discrepancy between training and inference time for the student model (Agarwal et al., 2024). MiniLLM (Gu et al., 2024), for instance, frames this as a reinforcement learning problem, optimizing reverse KL divergence over the fixed dataset. Lin et al. (2020). explored different distance metrics, including JSD and total variation distance. On the other hand, on-policy KD, pioneered by ImitKD (Lin et al., 2020), leverages samples generated by the student model itself. This approach addresses the training-inference mismatch by directly sampling target tokens from the student's own output. Inspired by on-policy imitating learning (Ross et al., 2011), Agarwal et al. (2024) introduced on-policy KD for language models, calculating both teacher and student token probabilities to train the student on its self-generated data. DistillM (Ko et al., 2024) further enhanced this by incorporating a mix of on-policy samples and ground truth. As discussed in §3.2, both supervised KD and on-policy KD are special cases of SKD. SKD outperforms both baselines and surpasses the performance from mixing ground-truth and on-policy samples.

**Speculative Decoding** (Leviathan et al., 2022; Chen et al., 2023) inspired our interleaved sampling approach in SKD. However, instead of sampling from teacher's distribution, SKD assesses the feasibility of student-proposed tokens within the teacher's top $K$ tokens. Interleaved sampling allows us to obtain high quality on-policy samples. Zhou et al. (2024); Liu et al. (2024) enhanced speculative decoding alignment with various KD methods. Building on these, we benchmarked SKD's speedup against KD baselines, finding it superior in both speed and token acceptance.

## 7 CONCLUSION

We propose Speculative Knowledge Distillation (SKD), a novel method that addresses the limitations of existing KD approaches. SKD leverages interleaved sampling to mitigate the discrepancy between training and inference samples and remove low-quality student-generated data. Our experiments consistently show SKD's superiority in both task-specific and task-agnostic distillations. SKD is robust to various model families, initialization, and dataset sizes. We also demonstrate its superiority in token-level mixing over sequence-level mixing. Furthermore, SKD-trained student models can accelerate speculative decoding, enabling new applications.

# 8 REPRODUCIBILITY STATEMENT

We discussed our model implementation, hyper-parameters, baseline details in Section 4. We include information about our dataset and evaluation metrics in Section 4.1. We discuss training and hyper-parameter details in Appendix D. We include examples of our prompt and response for all tasks in Appendix F. We include additional information about our evaluation metrics in Appendix G. All data, model and evaluation metrics are open sourced and available for research community. Upon the camera ready of the paper, we will include a link to a publicly accessible code repository, including all implementations about baseline KDs and SKD, running instructions and hyper-parameter settings. All results reported in this paper can be fully reproduced by the community.

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

## A   DIVERGENCE METRICS.

As previously discussed, for each token position, the auto-regressive language model (LM) $M$ can generate a probability distribution, $p(.|X, y_{<k})$. Assuming the teacher and student models share the same vocabulary $V$, we can directly employ divergence metrics, like KL divergence, to quantify the distance between their respective probability distributions. For token position $k$, let $P(C)$ and $Q(C)$ represent the vocabulary distributions of the teacher and student, respectively. The KL divergence at this position is calculated as $D_{KL}(P||Q) = \sum_{c \in C} P(c)(\log \frac{P(c)}{Q(c)})$. By using this divergence metric as our objective function, we can effectively conduct knowledge distillation between the teacher and student models.

## B   CAN SKD OUTPERFORM BASELINES UNDER DIFFERENT $K$ CHOICES?

In this section, we tested a wide range of $K$ values and showed SKD's performance across three text generation tasks. In Figure 6, we show that while extremely low or high $K$ values can not yield the most optimal performance, a broad range of $K$ values between 5 to 50 can consistently outperform supervised KD and on-policy KD across all three text generation tasks. The overall performance trend confirms our hypothesis that as $K$ increases, SKD degenerates to on-policy KD like behavior, while decreasing $K$ degenerates SKD to supervised KD like behavior. Consequently, neither the highest nor lowest values of $K$ are optimal.

We maintained a fixed $K$ of 25 (a common choice for top-K sampling) in our primary experiments to assess SKD's general effectiveness without extensive parameter tuning. Our results in this section suggest that further task-specific optimization of $K$ could potentially yield even better performance.

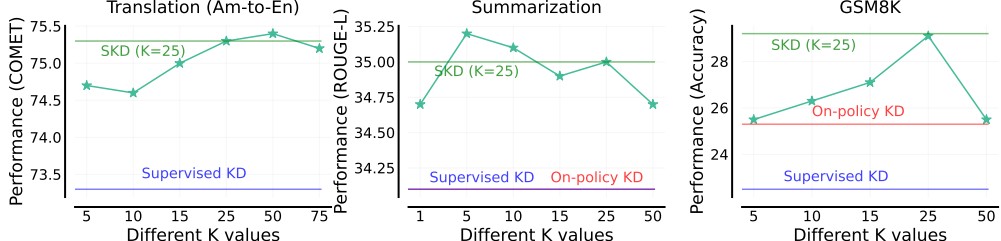

**Figure 6:** We test a wide range of $K$ values and show SKD's performance across three text generation tasks. We initialize student model with GEMMA-2B-IT. We use SKD with $K=25$ (a common choice for top-k sampling) for our experiments in prior sections. In this Figure, our results demonstrate that while extremely low or high $K$ values can not yield the most optimal performance, a broad range of $K$ values between 5 and 50 consistently outperforms supervised KD and on-policy KD across all three text generation tasks.

## C   ADAPTIVE K STUDY

We explored experiments by adjusting $K$ during training steps (decreasing $K$). However, we didn't obtain positive signals (it didn't beat constant $K$ approach). Our hypothesis is that SKD assumes that as the model improves, it naturally acts like on-policy KD. However, heuristically decreasing $K$ can force it to replace the token and behave like supervised KD, which introduces off-policy tokens, hurting the performance of the model. However, we encourage future works to explore more into this adaptive $K$ direction.

We also conducted experiments using adaptive $K$ strategy at decoding steps. Due to the autoregressive nature of our base models, token generation becomes more deterministic with longer sequences. This raises the question of whether reducing $K$ as the sequence grows could further improve performance. We experimented with a linear decay of $K$, decreasing it by for each additional generated token. We set a minimum $K$ to ensure our algorithm does not degenerate to supervised KD towards the end of decoding. In Table 4, we showed that adaptive SKD outperforms most of KD baselines across translation and summarization, it does not outperform constant SKD with K=25.

**Table 4:** In this Table, we showed that performance under adaptive $K$. We initialized $K$ with 50 and 25 respectively and $K$ is reduced linearly with increasing decoding step. We also setup clipping $K$ value so that adaptive SKD pipeline will not degenerate to supervised KD at later decoding stage. Cliiping value is set to be 25 for $k$ is initialized at 50 and 15 for $k$ is initialized at 25. Although adaptive SKD outperforms most of KD baselines across translation and summarization, it does not outperform constant SKD with $K$=25.

|  | Translation (Measured by COMET) | | | Summarization (Measured by ROUGE-L) | | |
|---|---|---|---|---|---|---|
|  | $K$ from 50 to 25 | $K$ from 25 to 15 | $K$=25 | $K$ from 50 to 25 | $K$ from 25 to 15 | $K$=25 |
| Performance | 75.1 | 75.0 | 75.3 | 25.2 | 28.7 | 29.1 |

We also encourage future work to study Top-$P$ criteria (Holtzman et al., 2020) for SKD pipeline. We believe that combing Top-$P$ and Top-$K$ criteria can lead to better performance of SKD pipeline.

## D  TRAINING AND HYPERPARAMETER DETAILS FOR BASELINES AND SKD

For both GEMMA-2B-IT and QWEN2-0.5B-IT models, we set the learning rate to 1e-5. We disabled dropout during sampling, as in direct preference learning (Rafailov et al., 2023). Maximum input and output lengths were set task-specifically: (256, 256, 1024, 1024) and (256, 512, 128, 1024) for translation, arithmetic reasoning, summarization, and math instruction, respectively. All baseline models used a batch size of 8 and a gradient accumulation step of 1. Student model is trained over 375, 375, 375 and 1225 training steps for summarization, GSM, math and translation tasks respectively. We train 7500 steps for full data training at Math. For extreme low data setting (with prompt size 100), we trained student model over 125 steps for summarization, GSM and translation tasks. We report checkpoint results with the lowest validation loss.

Initial experiments with top-p sampling (temperature=1, top-p=1) led to significantly degraded generation quality compared to greedy decoding (temperature=0). This was particularly evident in math instruction task, where accuracy halved. To balance exploration and performance, we gradually reduced temperature and top-p until top-p sampling matched or outperformed greedy decoding. We ultimately settled on temperature=0.5 and top-p=0.5. This setting effectively limits the answer space while allowing for sufficient exploration. To ensure fair comparison, we kept the student temperature and top-p parameters identical for on-policy KD and SKD.

Two additional hyperparameters, teacher model top-p and temperature, can influence SKD's performance. When a student token is rejected by the teacher, these parameters affect the resampled teacher token. Similar to prior work (Agarwal et al., 2024), we found that lower teacher temperatures improve distillation. We conducted a grid search over top-p $\{0.5, 1\}$ and temperature $\{0.2, 0.3, 0.4\}$ for translation, arithmetic reasoning, and summarization on respective validation sets. While the optimal temperature varied slightly across tasks, 0.2 generally performed well. Although a full vocabulary distribution (top-p=1) typically yielded the best results, reducing top-p to 0.5 improved ROUGE-L from 0.5 to 0.9 across different temperatures at Summarization task. Therefore, we set temperature to 0.2 for all tasks and top-p to 0.5 for summarization and 1 for other tasks. All hyperparameter settings are consistent between GEMMA-2B and QWEN2-0.5B-IT model.

In Section B, we explore a wide range of $K$ values and test SKD's performance across three text generation tasks. Our results demonstrate that a wide range of $K$ values can outperform baseline knowledge distillation methods, such as On-policy KD and Supervised KD. To ensure consistent evaluation, we selected a constant $K$ value of 25 (a common choice for Top-K sampling) for all tasks and ablation studies. However, future research could explore more dynamic $K$ selection strategies to potentially further enhance SKD performance.

**Table 5:** In this Table, we show performance of GEMMA-2B-IT (student), supervised FT GEMMA-2B-IT (student) and supervised FT GEMMA-7B-IT (teacher) at both task specific distillation and task agnostic distillation.

| | Task specific distillation | | | Task agnostic distillation | | | |
|---|---|---|---|---|---|---|---|
| | MT | Summ | GSM8K | Math | GSM$_{plus}$ | SVAMP | ASDiv |
| Baselines | COMET | ROUGE-L | Acc | Acc | Acc | Acc | Acc |
| GEMMA-2B-IT | 44.5 | 11.7 | 0 | 0.68 | 0.5 | 1.6 | 1.8 |
| Supervised FT GEMMA-2B-IT | 72.5 | 31.7 | 18.7 | 4.82 | 4.70 | 12.8 | 15.2 |
| Supervised FT GEMMA-7B-IT | 79.2 | 36.8 | 60.0 | 19.5 | 40.5 | 66.6 | 76.1 |

**Table 6:** In this Table, we show performance of QWEN-0.5B-IT (student), supervised FT QWEN-0.5B-IT (student) and supervised FT QWEN-7B-IT (teacher) at both task specific distillation and task agnostic distillation.

| | Task specific distillation | | | Task agnostic distillation | | | |
|---|---|---|---|---|---|---|---|
| | MT | Summ | GSM8K | Math | GSM$_{plus}$ | SVAMP | ASDiv |
| Baselines | COMET | ROUGE-L | Acc | Acc | Acc | Acc | Acc |
| Qwen-0.5B-it | 44.9 | 15.2 | 0 | 7.0 | 3.9 | 1.7 | 3.9 |
| Supervised FT QWEN-0.5B-IT | 57.6 | 29.2 | 31.8 | 2.6 | 3.9 | 10.5 | 14.1 |
| Supervised FT QWEN-7B-IT | 79.4 | 38.1 | 73.8 | 46.0 | 60.4 | 84.8 | 90.0 |

**Table 7:** This table presents baseline and SKD performance across four held-out test sets. The teacher model is a supervised FT GEMMA-7B, while the student model is an instruction-tuned GEMMA-2B. The left side of the table shows results for all baselines and SKD using 1,000 prompts from our constructed training set. The right side displays results using 10,000 prompts.

| | KD initiates with GEMMA-2B-IT (1k prompts) | | | | KD initiates with GEMMA-2B-IT (10k prompts) | | | |
|---|---|---|---|---|---|---|---|---|
| | Math | GSM$_{plus}$ | SVAMP | ASDiv | Math | GSM$_{plus}$ | SVAMP | ASDiv |
| Baselines | Acc | Acc | Acc | Acc | Acc | Acc | Acc | Acc |
| SFT | 4.82 | 4.70 | 12.8 | 15.2 | **10.9** | 16.8 | 42.3 | 56.8 |
| Supervised KD | 7.92 | 12.1 | 42.7 | 53.4 | 10.7 | 16.1 | 46.7 | 63.3 |
| On-policy KD | 7.92 | 15.1 | 44.5 | 57.4 | 10.3 | 18.6 | 49.9 | 61.8 |
| SKD | **9.56** | **16.9** | **46.9** | **59.1** | 10.2 | **19.7** | **50.4** | **68.4** |

# E  ADDITIONAL RESULTS

Tables 5 and 6 show the performance of GEMMA-2B-IT (student), QWEN-0.5B-IT (students) and supervised FT teacher models (GEMMA-7B-IT and QWEN-7B-IT) under task-specific and task-agnostic distillation. Table 7 presents task-agnostic instruction tuning results for baseline and SKD models across four held-out test sets, using both low-data (1k prompts) and full-data (10k prompts) training regimes. These results align with those shown in Figure 3.

# F  TASK INPUT AND OUTPUT FORMAT

We include input and output formats for four text generation tasks that we studied in this paper: math instruction following (Table 8), arithmetic reasoning (Table 9), dialogue summarization (Table 10) and translation (Table 11).

---

**Prompt for model:**

Solve the following math problem step-by-step.
Simplify your answer as much as possible. Present your final answer as boxed {Your Answer}.
For a positive integer $n$, let $H_n = 1 + \frac{1}{2} + \frac{1}{3} + \cdots + \frac{1}{n}$ Compute $\sum_{n=1}^{\infty} \frac{1}{(n+1)H_n H_{n+1}}$.

**Output for model:**

Step 1: First, let's understand what the problem is asking for. We are given the expression $H_n = 1 + \frac{1}{2} + \frac{1}{3} + \cdots + \frac{1}{n}$ for a positive integer n, and we need to compute the sum $\sum_{n=1}^{\infty} \frac{1}{(n+1)H_n H_{n+1}}$.

Step 2: To begin solving this problem, we can start by manipulating $\frac{1}{(n+1)H_n H_{n+1}}$ to see if we can simplify it in any way. By doing this, we can rewrite $\frac{1}{(n+1)H_n H_{n+1}}$ as $\frac{H_{n+1} - H_n}{H_n H_{n+1}}$.

Step 3: This allows us to express the given sum as a telescoping series, where each term of the sum cancels out most of the preceding and following terms.

Step 4: Taking the sum $\sum_{n=1}^{\infty} \frac{1}{(n+1)H_n H_{n+1}}$ in parts, we can break it down term by term to simplify the expression and find the pattern of the telescoping series.

Step 5: By expressing each term of the sum as a difference involving the harmonic numbers $H_n$, we can observe how the terms cancel out, and this leads us to the result of the sum.

Step 6: To validate and confirm the result, we need to acknowledge the fact that $H_n \to \infty$ as $n \to \infty$, and we can prove this by showing that the sum of the harmonic series diverges.

Step 7: This proof involves demonstrating that the sum of the harmonic series increases without bound as n increases, which verifies the behavior of $H_n$ as $n \to \infty$.

Step 8: Therefore, the final answer to the given problem is $\boxed{1}$, and it is backed by the property of harmonic numbers tending to infinity as the terms increase.

Answer:

$\boxed{1}$

---

**Table 8:** This outlines the input and output format for math instruction following task. The output includes chain-of-thought steps leading to the computed answer, which is presented in a boxed format. We evaluate these answers by comparing their numerical equivalence to the ground truth.

---

**Prompt for model:**

Q: American carmakers produce 5 650 000 cars each year. Then they distribute it to 5 car suppliers. The first supplier receives 1 000 000 cars each. The second supplier receives 500 000 cars more while the third car supplier receives the same number as the first and second suppliers combined. The fourth and the fifth supplier receive an equal number of cars. How many cars do the fourth and fourth suppliers each receive?

**Output for model:**

The second supplier receives 1 000 000 + 500 000 = 1 500 000 cars.

The third supplier receives 1 000 000 + 1 500 000 = 2 500 000 cars.

So the first to third suppliers receive a total of 1 000 000 + 1 500 000 + 2 500 000 = 5 000 000 cars.

Thus, 5 650 000 - 5 000 000 = 650 000 cars are left to be divided by the two other suppliers.

Therefore, the fourth and fifth suppliers receive 650 000/2 = 325 000 cars each.

#### 325,000

---

**Table 9:** This outlines the input and output format for arithmetic reasoning task. The output includes chain-of-thought steps leading to the computed answer, which is presented after "####". We evaluate these answers by comparing their numerical equivalence to the ground truth.

---

**Prompt for model:**

#Person1#: Hi, Mr. Smith. I'm Doctor Hawkins. Why are you here today?

#Person2#: I found it would be a good idea to get a check-up.

#Person1#: Yes, well, you haven't had one for 5 years. You should have one every year.

#Person2#: I know. I figure as long as there is nothing wrong, why go see the doctor?

#Person1#: Well, the best way to avoid serious illnesses is to find out about them early. So try to come at least once a year for your own good.

#Person2#: Ok.

#Person1#: Let me see here. Your eyes and ears look fine. Take a deep breath, please. Do you smoke, Mr. Smith?

#Person2#: Yes.

#Person1#: Smoking is the leading cause of lung cancer and heart disease, you know. You really should quit.

#Person2#: I've tried hundreds of times, but I just can't seem to kick the habit.

#Person1#: Well, we have classes and some medications that might help. I'll give you more information before you leave.

#Person2#: Ok, thanks doctor.

**Output for model:**

Mr. Smith's getting a check-up, and Doctor Hawkins advises him to have one every year. Hawkins'll give some information about their classes and medications to help Mr. Smith quit smoking.

---

**Table 10:** This outlines the input and output format for dialogue summarization task. The output is the summary of above dialogue. We evaluate summary by computing its ROUGE-L score to the ground truth.

---

**Prompt for model:**

Translate Assamese sentence into English. This is the Assamese sentence: [Assamese Texts]. Here is the English translation:

**Output for model:**

It was to last for the next 40 years and would be fought for real, by proxy armies, on battlefields from Africa to Asia, in Afghanistan, Cuba and many other places.

---

**Table 11:** This outlines the input and output format for translation task. The output is the English translation. We evaluate translation by computing its COMET score to the reference and source.

## G    EVALUATION METRICS

As in (Freitag et al., 2022), we adopted the learned evaluation metric COMET (Rei et al., 2020) for assessing our translation outputs, instead of the n-gram-based BLEU (Papineni et al., 2002). Specifically, we used the comet-da-22 checkpoint (Rei et al., 2022), a widely recognized metric that exhibits strong correlation with human judgments at WMT22 (Freitag et al., 2022). COMET is a learned metric trained on a large-scale dataset of human judgments for evaluating translation quality. It takes source, translation, and reference as input, producing a quality score between $0$ and $1$. A COMET score improvement of $0.5$ to $1.0$ indicates a substantial improvement.

Tables 8 and 9 illustrate the structured output format used to derive final answers from math instruction tasks and GSM-8k. We employed the evaluation code from (Yuan et al., 2024) to assess the mathematical equivalence of parsed answers and ground truths. This involved converting mathematical symbols like $\pi$ into their numerical values for more accurate numerical comparisons, rather than relying solely on string matching. Implementation details can be found in the original paper (Yuan et al., 2024). We reported accuracy for both tasks.

We evaluated summarization output against references using ROUGE-L (Lin, 2004). ROUGE-L ranges from 0 to 100, with a 1-point increase indicating substantial improvement.

---

**Prompt for model:**

Translate Assamese sentence into English. This is the Assamese sentence: [Assamese Texts]. Here is the English translation:

**Reference translation:**

Italian is also the everyday language used by most of those who work in the state while Latin is often used in religious ceremonies.

**On-policy student sampled output (COMET: 53):**

The Estonian language is also used in the Estonian-speaking countries as a national language, and is used in the Estonian-speaking countries as a medium of instruction in the Estonian-speaking countries.

**SKD student sampled output (COMET: 80):**

The Latin language is still used in the country to a large extent, though it is not the official language.

---

**Table 12:** We sample training data from SKD and on-policy methods. Our results show that solely sampling from the student model can lead to hallucinated translations and limited training signals (On-policy output). Conversely, SKD generates fluent, semantically similar outputs while staying close to the student's distribution (SKD output). Student model is supervised FT QWEN-0.5B.

## H  CASE STUDY BETWEEN ON-POLICY AND SKD SAMPLES

Tables 12 and 13 illustrate low-quality training examples generated by on-policy models. In Table 12, the student model hallucinates "Estonian language," a nonexistent term in the source, leading to a poor translation. SKD with teacher's interleaved sampling avoids this issue. In Table 13, on-policy KD samples repeat the n-gram "in the past," producing nonsensical translations. SKD's interleaved sampling helps prevent this pathology. In Table 14, we sampled training data from SKD and on-policy methods using the GEMMA-2B-IT model. These samples were collected during the early stages of training. The on-policy sample exhibited repetitive outputs and errors in arithmetic calculations. While our SKD model also produced incorrect answers, incorporating teacher's corrections enabled it to perform some intermediate steps correctly and maintain accurate arithmetic calculations. In Table 15, we sampled training data from SKD and on-policy methods using the Gemma-2B-it model. These samples were collected during the late stages of training. We observed that teacher model can perform some intermediate correction on the reasoning step, ultimately leading to correct answer.

---

**Prompt for model:**

Translate Assamese sentence into English. This is the Assamese sentence: [Assamese Texts]. Here is the English translation:

**Reference translation:**

It was to last for the next 40 years and would be fought for real, by proxy armies, on battlefields from Africa to Asia, in Afghanistan, Cuba and many other places.

**On-policy student sampled output (COMET: 36):**

This has been a long war, and has been fought in the past, in the past, in the past, in the past, in the past, in the past, in the past, in the past, in the past, in the past, in the past, in the past, in the past, in the past, in the past, in the past, in the past, in the past, in the past, in the past, in the past, in the past, in the past, in the past, in the past, in the past, in the past, in the past, in the past, in the past, in the past, in the past, in the past, in the past, in the past, in the past, in the past, in the past, in the past, in the past, in the past, in the past, in the past, in the past, in the past, in the past, in the past, in the past, in the past, in the past, in the past, in the past, in the past, in the past, in the past, in the past, in the past, in the past, in the past, in the past, in the past, in the past, in the past, in the past, in the past, in the past, in the past, in the past, in the past, in the past, in the past, in the past, in the past, in

**SKD student sampled output (COMET: 72):**

This has been going on for 40 years and is still ongoing, with the possibility of war, a civil war, Afghans in the Balkans, Ethiopia, Kenya, Somalia, and possibly even the United States.

---

**Table 13:** We sample training data from SKD and on-policy methods. Our results show that solely sampling from the student model can lead to poor translations and limited training signals (On-policy output). Conversely, SKD generates fluent, semantically similar outputs while staying close to the student's distribution (SKD output). We omit source sentence due to font format issues. Student model is supervised FT QWEN-0.5B.

---

**Prompt for model:**

Q: Libby is building an igloo in her backyard using bricks of snow. She builds her igloo in rows, using a total of 10 rows of bricks of snow. The bottom half of the igloo has 12 bricks of snow in each row while the top half of the igloo has 8 bricks of snow in each row. How many bricks of snow did Libby use for her igloo? A:
**Correct Answer:**

In the bottom half of the igloo, there are 10 rows / 2 = 5 rows of bricks of snow.

So in the bottom half of the igloo, there are 5 rows * 12 bricks of snow in each row = 60 bricks of snow.

In the top half of the igloo, there are 5 rows * 8 bricks of snow in each row = 5*8=4040 bricks of snow.

There is therefore a total of 60 + 40 = 100 bricks of snow in the igloo. #### 100

**On-policy student sampled output:**

$35-$160= $125
Libby has $160-$125=$35
Libby has 160-35= $125
Libby has 160-35= $125...

**SKD student sampled output:**

The bottom half of the igloo has 12 x 10 = 120 bricks of snow.

The top half of the igloo has 8 x 10 = 80 bricks of snow.

Libby used 120 + 80=200 bricks of snow. #### 200

---

**Table 14:** We sample training data from SKD and on-policy methods using the GEMMA-2B-IT model. These samples are collected during the early stages of training. The on-policy sample exhibit repetitive outputs and errors in arithmetic calculations. While our SKD model also produces incorrect answers, incorporating teacher's corrections enables it to perform some intermediate steps correctly and maintain accurate arithmetic calculations.

---

**Prompt for model:**

Q: John Smith buys 3 cakes for $12 each and splits the cost with his brother. How much did he pay? A:
**Correct Answer:**

He spent 3*12=36 total So he paid 36/2=18 #### 18

**On-policy student sampled output:**

Each brother pays $12/3 = 4 each

So he pays $12 - $4 = 8 #### 8

**SKD student sampled output:**

The 3 cakes cost John 3 x $12 = $36

So he paid $36 / 2 = $18 each #### 18

---

**Table 15:** We sample training data from SKD and on-policy methods using the GEMMA-2B-IT MODEL. These samples are collected during the late stages of training. We observe that teacher model can perform some intermediate correction on the reasoning step, ultimately leading to correct answer.

# I OVER-FITTING AT SFT STAGE CAN CAUSE SUB-OPTIMAL PERFORMANCE AT POST-TRAINING KDS

Figure 7 demonstrates that overfitting during the SFT stage can hinder optimal performance in post-training KD. **End-to-end KD methods, such as SKD, which can train from arbitrary base model and bypass the SFT stage, offer significant benefits**. To investigate this, we selected three checkpoints from the supervised FT stage of the student model, corresponding to training steps 64, 128, and 192. Based on validation, the checkpoint at step 128 represents the converged model, while step 192 exhibits signs of overfitting (Figure a). Figure (b) shows that the converged checkpoint (step 128) outperforms the overfitted checkpoint (step 192). Moreover, we found that training from an earlier checkpoint (step 64) can yield better results than the overfitted checkpoint (step 192). This observation is consistent for both on-policy and SKD. However, SKD is an end-to-end training pipeline. Our findings confirm that overfitting during the SFT stage can lead to suboptimal performance in post-training KD. As shown in Section 5.4, training with extremely limited data can inevitably lead to overfitting during the SFT stage, further highlighting the superiority of our end-to-end SKD pipeline compared to two-stage KD approaches like on-policy KD.

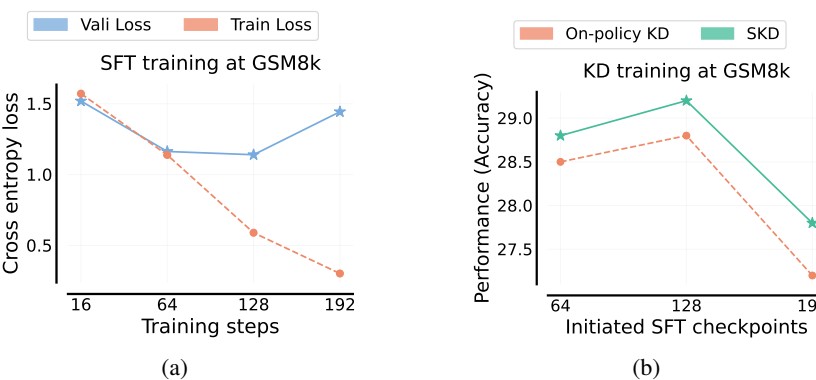

(a)                                                      (b)

**Figure 7:** In this Figure, we demonstrate that over-fitting during the SFT stage can hinder optimal performance in post-training knowledge distillation. **End-to-end KD methods, such as SKD, which can train directly from arbitrary base model and bypass the SFT stage, offer significant benefits**. Figure (a) presents three checkpoints from the SFT stage, corresponding to training steps 64, 128, and 192. Based on validation loss, the checkpoint at step 128 represents the converged model, while step 192 shows signs of overfitting. Figure (b) demonstrates that the converged checkpoint (step 128) outperforms the overfitted checkpoint (step 192). This finding is consistent for both on-policy KD and SKD.

# J TRAINING DETAILS OF MIXED TRAINING OF SUPERVISED KD AND ON-POLICY KD

The parameter setting for this baseline is identical for the setup in Appendix D. We start to train student model with supervised KD samples for 188 steps (roughly 1.5 epoches), then we perform on-policy KD for another 187 steps. Total training step is 375. Initial learning rate is $1e$-5 and linear decay is applied during training process.

# K SPEEDUP TO SPECULATIVE DECODING

Motivated by (Zhou et al., 2024), we further investigate the speedup achieved by each baseline model when used as a draft model to approximate a teacher model during speculative decoding (Leviathan et al., 2022)]. As depicted in Figure 8, SKD-trained student models consistently exhibit the highest speedup and token acceptance ratio (Approximation model will accept or reject tokens depending on whether draft token is sampled under its probability distribution) across translation and summarization tasks. These models accelerate instruction-tuned GEMMA-2B by 1.21x and 1.17x in translation and summarization, respectively, while also improving token acceptance ratio by 1.71x

and 1.85x. These results demonstrate that SKD not only produces superior performance but also generates the closest distribution to the teacher model. This opens up a new application: utilizing SKD to train powerful draft models and accelerate the speculative decoding process.

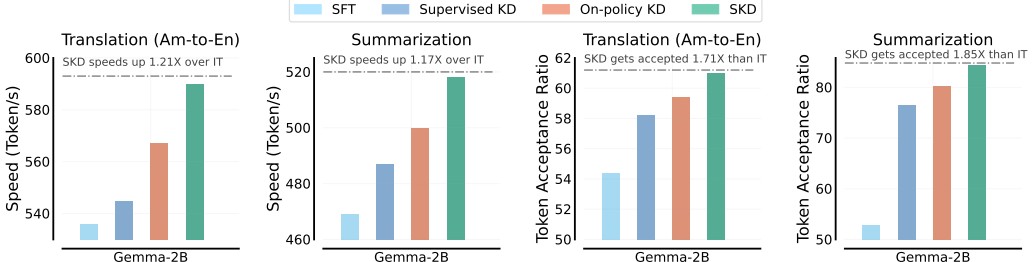

**Figure 8:** We utilize student models trained with various baseline KD methods as draft models and a supervised fine-tuned GEMMA-7B-IT model as the approximation model for speculative decoding. Our results demonstrate that SKD-trained student models significantly outperform the others, achieving the highest speedup and token acceptance ratio. SKD accelerates GEMMA-2B-IT by 1.21x and 1.17x in translation and summarization, respectively, while also improving token acceptance ratio by 1.71x and 1.85x.

## L    REJECTION RATE ANALYSIS

We calculate the rejection rate of the teacher model across different student initializations and training steps. The rejection rate is defined as the total number of tokens re-sampled by the teacher divided by the total number of tokens generated by both the teacher and student. After 1,225 training steps, the rejection rate decreases to 5.4% for the instruction-tuned Gemma-2B and 0.1% for the supervised fine-tuned (SFT) Gemma-2B, respectively.

We randomly select 100 samples from the Assamese-to-English training dataset. Figure 9 illustrates the estimated rejection rates of the teacher model (Supervised Fine-Tuned Gemma-7B) with two student models: Gemma-2B-IT and Gemma-2B-SFT. For Gemma-2B-IT (left panel), the rejection rate starts high, with 40.9% of tokens requiring teacher resampling during sequence generation. This rate drops sharply within the first 150 training steps, from 40.9% to 9.7%, and continues to decline gradually as training progresses. This trend supports our hypothesis that, in early training stages, SKD behaves more like supervised knowledge distillation (with  40.9% of tokens replaced by the teacher). As training advances, SKD transitions to a more on-policy approach, with only  5% of tokens replaced by the teacher.

In the right panel of Figure 9, we evaluate the rejection rate with the SFTed Gemma-2B student model. The initial rejection rate is significantly lower (4.7%), reflecting the student model's stronger initialization and closer proximity to the teacher. The rejection rate continues to decrease as training progresses, maintaining a consistent downward trend. Despite the lower initial rejection rate and the more on-policy behavior of SKD with the SFTed student model, we demonstrate SKD's superior performance in Table 2.

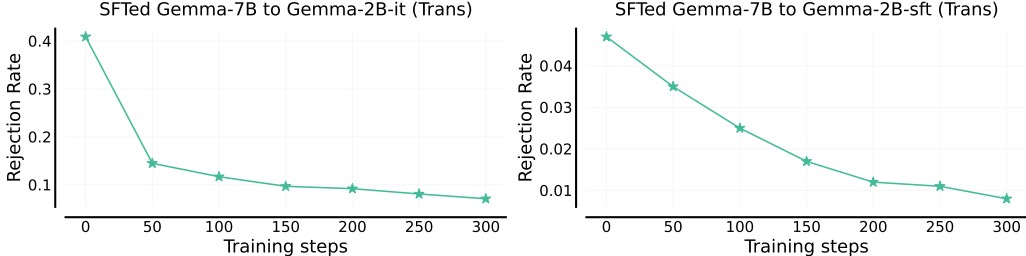

**Figure 9:** We compute the rejection rate of the teacher model under different student initializations with various training steps at Assamese to English translation task. We estimated rejection rates of teacher model supervised FT Gemma-7B with student models gemma-2b-it and gemma-2b-sft, respectively. We found that the rejection rate has a sharp decrease in the first 150 iterations. Student model with higher quality initialization (SFTed) can achieve much lower rejection rate.

## M    COMPUTATIONAL ANALYSIS

An important consideration in SKD is the computational cost associated with on-policy sampling. Using the estimated rejection rates as a reference, we can analyze the computational costs of this approach. By leveraging the isoFLOP cost calculation introduced in (Kaplan et al., 2020), we estimate the inference cost of our models using Equation 2:

$$Computation = 2ND + 2N\sqrt{D} * L \tag{2}$$

where $N$ is the number of parameters. $D$ is the number of input tokens and $L$ is the number of decoding steps.

Based on our rejection rate analysis, we compute the expected rejection rate for Gemma-2B-IT during training. For simplicity, we assume an expected rejection rate of 10% (slightly higher than the actual rate during training but convenient for analysis), corresponding to an expected acceptance rate of 90%. We estimate the expected number of generated tokens using a simplified version of Algorithm 1 shows a simplified algorithm where $\gamma = 1$. In practice, we use $\gamma = 5$.

Using a geometric series approach similar to (Leviathan et al., 2022), we estimate the expected number of student-generated tokens with Equation 3:

$$E(\# \ student \ generated \ tokens) = \frac{1 - \alpha^{\gamma+1}}{1 - \alpha} \tag{3}$$

where $\alpha$ is the expected acceptance ratio and $\gamma$ is the sequence length generated by the student model per run. With an expected acceptance ratio of $0.9$, the expected number of student-generated tokens per run is $(1 - 0.9^5)/(1 - 0.9) = 4.1$. For simplicity, we round this to $4$. For 5 student-proposed tokens, the teacher evaluates all tokens in parallel and typically rejects the last one. Thus, we can approximate that in SKD, the student generates 80% of the tokens, while the teacher generates the remaining 20%.

To quantify this in terms of inference cost, we assume $N_{student}$=2B, $N_{teacher}$=7B, $D$=256, $L$=200. If the ground truth is unavailable and must be sampled from the teacher, supervised KD is roughly 3.5x higher than on-policy KD, while the sampling cost of SKD is approximately 1.72x higher than on-policy KD. SKD can thus save approximately 50% of the computational cost compared to directly sampling from the teacher. This calculation is based on an instruction-tuned student model; using an SFTed student model would yield additional computational savings. This example provides a high-level overview of the computational requirements for SKD sampling.

## N    VALIDATION CONVERGENCE ACROSS METHODS

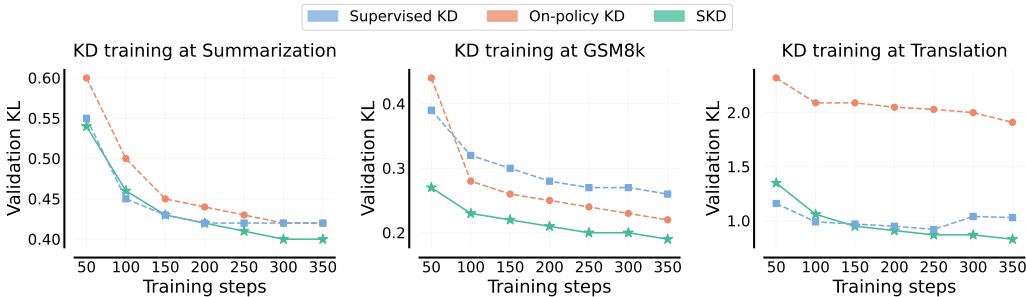

**Figure 10:** In this Figure, we report the validation loss of various baseline KD strategies across 350 training steps, distilling Gemma-7B to Gemma-2B-it model. Our findings demonstrate that SKD consistently achieves the lowest validation loss by the final iteration across all three tasks.

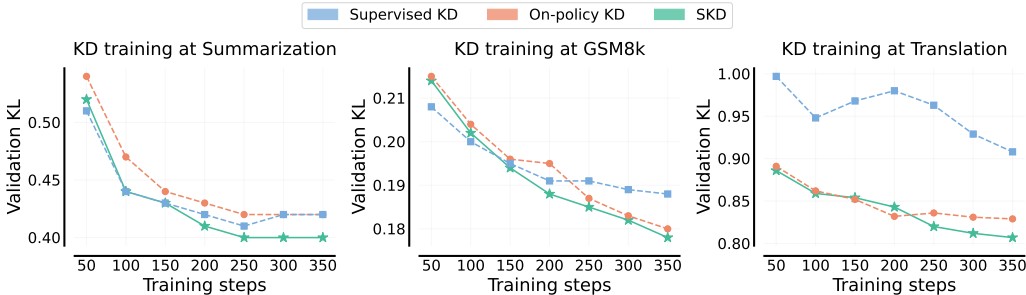

**Figure 11:** We report the validation loss of various baseline KD strategies across 350 training steps, distilling Gemma-7B to SFTed Gemma-2B model. Our findings demonstrate that SKD consistently achieves the lowest validation loss by the final iteration across all three tasks.

In Figure 10, we present the validation loss of various baseline KD strategies over 350 training steps, where Gemma-7B is distilled into the Gemma-2B-IT model. Our results show that SKD consistently achieves the lowest validation loss by the final iteration across all three tasks. A key observation is that the lower quality of on-policy samples can lead to slower initial convergence. This is evident at step 50, where on-policy KD exhibits higher validation losses compared to both supervised KD and SKD across all tasks. In translation tasks, student models trained with on-policy KD continue to show higher validation losses than those trained with SKD or supervised KD, reflecting the inferior quality of on-policy samples, as indicated by the lower COMET scores in Table **??**. For simpler tasks, such as translation and summarization, supervised KD achieves lower initial validation losses compared to SKD and on-policy KD. However, SKD rapidly improves in later iterations, ultimately achieving the lowest validation loss by the end of training, while supervised KD risks overfitting, as shown in the rightmost panel of Figure 10. For more complex tasks, such as the GSM task, SKD's interleaved sampling method enables faster convergence and consistently lower validation losses throughout training, outperforming both on-policy and supervised KD.

In Figure 11, we report the validation loss of baseline KD strategies over 350 training steps, distilling Gemma-7B into the SFTed Gemma-2B model. SKD consistently achieves the lowest validation loss by the final iteration across all three tasks. Additionally, we observe that as the base quality of the student model improves, on-policy KD achieves lower validation losses compared to supervised KD by the end of training. This suggests that on-policy approaches benefit from improved sample quality. Most importantly, SKD achieves the best convergence within 350 training steps for both model initializations, outperforming both supervised KD and on-policy KD.

The training process remains stable across on-policy KD, supervised KD, and SKD, as the KL loss provides a stable training objective. The primary difference between these approaches lies in the effectiveness of the samples for learning, as measured by their ability to minimize validation loss. Through a comparative analysis of validation losses for both instruction-tuned (IT) and supervised fine-tuned (SFT) student model initializations, we demonstrate that SKD achieves superior convergence, consistently attaining the lowest validation loss compared to baseline methods.

## O QUANTIFY SAMPLE OOD ISSUES WITH DIFFERENT KD APPROACHES

To quantify the out-of-distribution (OOD) issues in the samples, we computed the per-sample perplexity (PPL) using both the teacher model (supervised fine-tuned Gemma-7B) and the student model (instruction-tuned Gemma-2B). Our analysis included ground truth data, on-policy samples generated by the student, and samples produced by SKD. We randomly selected 100 prompts from the translation training dataset for student and SKD sampling, while for supervised KD, we used ground truth outputs.

As shown in Table 16, the teacher model assigned a significantly higher PPL to student-proposed samples 46.8, compared to ground truth samples 1.57, indicating a substantial divergence between the student's proposals and the teacher's distribution. In contrast, SKD-generated samples achieved a much lower PPL of 10.8, suggesting that the teacher model can more accurately evaluate the quality of SKD samples compared to those proposed by the student.

**Table 16:** In this Table, we quantify sample out-of-distribution (OOD) issues by estimating the per-sample perplexity (PPL), using both the teacher (supervised FT Gemma-7B) and the student (instruction-tuned Gemma-2B). Our analysis included ground truth data (supervised KD), on-policy samples proposed by the student, and samples generated by SKD's interleaved sampling. Our finding demonstrates that both student proposed samples and ground truth are OOD to teacher and student model respectively. With large drops at PPL value, we demonstrate that SKD's interleaved sampling can mitigate sample OOD issues and lead to more effective KD process.

|  | PPL from SFTed Gemma-7B (Teacher) | | | PPL from Gemma-2B-it (Student) | | |
|---|---|---|---|---|---|---|
|  | On-policy | SKD | Supervised KD | On-policy | SKD | Supervised KD |
| PPL | 46.8 | 10.8 | 1.57 | 44.0 | 358 | 5606.3 |

Table 16 further reveals that ground truth samples exhibit a high PPL of 5606.3 under the student model, confirming their off-policy nature. Importantly, our SKD approach, which combines on-policy token proposals with teacher corrections, significantly reduces the PPL to 358 compared to ground truth samples. This demonstrates that SKD samples are better aligned with the student model's distribution, enabling more effective on-policy learning.

