# OpenReview forum: "Speculative Knowledge Distillation: Bridging the Teacher-Student Gap Through Interleaved Sampling"
_ICLR.cc/2025/Conference — ICLR 2025 Poster_

### Official Review · Reviewer_EjjZ · 2024-11-02

**Soundness:** 4
**Presentation:** 4
**Contribution:** 3
**Rating:** 6
**Confidence:** 2

**Summary:**

The method seems very simple and, to some extent, sounds like an interpolation between two distillation strategies. Specifically, in Supervised KD the next-token distribution of student and teacher are trained to be similar on a fixed text, while in On-policy it is done on student’s generated text. The method thus proposes to evaluate the tokens of the student and see if it’s within TopK of the teacher’s, if not, the generated tokens of the student is discarded and the rest is followed with teacher’s tokens. But if the generated token satisfies the TopK check, we continue with student’s tokens.


The idea is clearly motivated in the paper and makes sense. The quantitative results also seem promising (across tasks the method is mostly outperforming prior work by a good margin). Especially the method is consistently outperfoming On Policy KD, which supports the proposed idea of switching to teacher tokens if the student tokens are bad.

**Strengths:**

**(a)** I think the paper is extremely well written. Especially for someone who might not be fully familiar with how Knowledge Distillation is applied for the LLMs, I really appreciate how clearly the authors explain prior work and how they relate to each other. With only the first pass, I was able to grasp the ideas across related work and the main contribution of the work.


**(b)** I think the simplicity of method is one of its advantages. Simply checking the student token's quality, based on whether it's within TopK of the teacher's seems simple and intuitive. (see also Question a).


**(c)** The quantitative results, especially compared to On Policy KD, seem promising (please see the disclaimer in Weaknesses section).

**Weaknesses:**

**(a)** I like the idea of evaluating the student’s suggestions by the teacher and selecting the Top-K based on the teacher’s probabilities. However, what concerns me is the computational cost of the method. Given that this is the main contribution of the work, I think it has to be more clearly studied. Specifically, what I really like to see is how often the line 5 in Algorithm 1 is triggered. In Lines 212-216 it is essentially claimed that this trigger rate is decreased over training time, I think it has to be evidenced with a figure (x axis being the training steps and y axis being the trigger rate). In a similar sense, I think it should be reported what is the final rate (i.e how likely it is for the student’s tokens to be in TopK of the teacher’s after the distillation).


**(b)** Moreover, I think the value for K should be studied. Essentially for high Ks the method should get somewhat similar to On Policy KD and for lower ones should be closer to Supervised KD? So there should be a sweet spot. Also I’m wondering if the value for K needs to be adjusted based on the training step? In other words, maybe higher or lower Ks would be better at the early or later stages of distillation? Some ablation experiments (with different numbers for K) could be helpful in addressing this.

------ Minor issues ------

**(c)** In Line 119, it is stated that “the temperature t `introduces` randomness”. I think this is not entirely correct as it’s the sampling that introduces randomness. The temperature adjusts the distribution so it `adjusts` the randomness (as it is also stated correctly in the following sentence).

**(d)** In Line 184, “Correct previous mistakes”,  it is not clear what `mistake` is referring to. Is a mistake a poorly generated Token? What does `correcting` refer to?

**(e)** Line 250: “Since our assumption `is` that”

**(f)** In the pseudo-code (Algorithm 1), some variables, such as alpha in line 3, are not defined.

**(g)** This is not really an issue, rather a suggestion. A tiny figure to explain concepts defined in Section 3.1 would be very helpful. Basically a 1-row figure divided into maybe 4-5 sub-columns, with every column showing how each of the prior work applies the loss at token-level. Maybe something like Figure 1 in https://arxiv.org/pdf/2104.09044 I think Figure (2) can be refactored a bit so that it also includes other prior work.




------ Disclaimer ------

I’m not very familiar with KD approaches for LLM, and therefore it is possible that I haven't noticed a missing baseline. I would really appreciate it if other reviewers could also verify if the set baselines is complete.

**Questions:**

**(a)** It is not 100% clear to me why would one replace a `bad token` with a token sampled from teacher's distribution. Could the authors give some intuition of why not re-attempt the sampling from student's distribution (line 4 in Algorithm 1) to replace the last token but under the condition that it's within TopK of the teacher? In otherwords, maybe the `bad` token sampled from student's distribution can be replaced with a `better` token that is sampled from the same distribution, rather than one that comes from the teacher's. If it is within the computational budget, an experiment to demonstrate this could also help prove the point.


**(b)** If I understood correctly, your method does not add any computational overhead to the distillation process, as it is a simple probability check and a token replacement. If so, please assert this both in rebuttal and also in the paper. Otherwise, a clear discussion on computational overhead is missing.

---

> ### Author Response · Authors · 2024-11-21
> **Response to Reviewer EjjZ regarding to weakness a) Part 1**
>
> Corresponding to "(a) I like the idea of evaluating the student’s suggestions by the teacher and selecting the Top-K based on the teacher’s probabilities. However, what concerns me is the computational cost of the method. Given that this is the main contribution of the work, I think it has to be more clearly studied. Specifically, what I really like to see is how often the line 5 in Algorithm 1 is triggered. In Lines 212-216 it is essentially claimed that this trigger rate is decreased over training time, I think it has to be evidenced with a figure (x axis being the training steps and y axis being the trigger rate). In a similar sense, I think it should be reported what is the final rate (i.e how likely it is for the student’s tokens to be in TopK of the teacher’s after the distillation)."
>
> Thanks for the comment! We provide an estimate below, and will include details in the final version.
>
> The following Table is Table
>
> |From Gemma-2b-it |  0  |  50  |  100  |  150  | 200  | 250  | 300  | 750  | 1000  | 1225  |
> |----------------|--------|----------------|----------------|----------------|----------------|----------------|----------------|----------------|----------------|----------------|
> |   SKD-25  |  0.409  |      0.145      |      0.117      |      0.097      |      0.092     |     0.081     |     0.071     |    0.058     |     0.055     |     0.054     |
>
> The following Table is Table 2
>
> |From SFT Gemma-2b |  0  |  50  |  100  |  150  | 200  | 250  | 300  | 750  | 1000  | 1225  |
> |----------------|--------|----------------|----------------|----------------|----------------|----------------|----------------|----------------|----------------|----------------|
> |   SKD-25  |  0.047  |      0.035      |      0.025      |      0.017      |      0.012     |     0.011     |     0.008     |    0.002    |     0.001    |     0.001     |
>
> Following the suggestion, we compute the rejection rate (rejection rate is defined as the total number of tokens that are re-sampled by teacher divided by the total number of tokens generated by both teacher and student) of the teacher model under different student initializations with training steps. We randomly select 100 samples from the training data. We estimated rejection rates of teacher model Supervised FT Gemma-7B with student models gemma-2b-it and gemma-2b-sft. For gemma-2b-it as a student, rejection rate was initially high (around 40.9% teacher resample during one sequence generation). It dramatically reduced at the first 150 steps from 40.9% to 9.7%. It gradually drops as training goes further. This validates our hypothesis that at early training iterations, SKD behaves more supervised KD-like (around 40.9% tokens are replaced by teacher) and as training goes on towards the end of training, SKD will behave more on-policy like with only 5% tokens will be replaced.
>
> We further estimate rejection rate with the SFTed gemma-2b student model. We find out with relatively better student model initializations (also higher proximity to teacher). Rejection rate can be quite low at the beginning (around 4.7%). The trend stays the same as the rejection rate continues to decrease as the training step increases. Despite relatively low rejection rate for SKD under SFTed student model, we still demonstrate SKD’s superior performance against on-policy KD at Table 2.

---

> > ### Comment · Reviewer_EjjZ · 2024-11-23
> > **Concerns are Adressed**
> >
> > I thank the authors for their detailed reply and my concerns are now addressed.
> >
> > Regarding weakness (a), I think the provided tables on the rejection rate are helpful and clearly support the claims of the paper. Therefore, please add these at least to the supplement and clearly refer to them in the main paper.
> >
> > Regarding weakness (b), it seems that this was an oversight from my side, as it was already discussed in the paper. I thank the authors for addressing this as well.
> >
> > Regarding the computational overhead w.r.t on-policy (and similarly gains over SupervisedKD), I think it would be easier if these also appear in supplement. I think however, given that it's not very straight forward to calculate the overhead (as it is based on the rejection rate, the K, etc.) I think empirical result (overall distillation time) would be easier for future readers to grasp.

---

> > > ### Author Response · Authors · 2024-11-23
> > > **Thanks a lot for your reply!**
> > >
> > > Thanks a lot for your reply! I am glad that you feel more confident about our work! We will definitely include all empirical and theoretical time complexity in the Appendix. All other revisions will be done in the paper as well.

---

> ### Author Response · Authors · 2024-11-21
> **Response to Reviewer EjjZ regarding to weakness a) Part 2**
>
> With the estimated rejection rates as reference, we can further analyze computational costs of this approach. Leveraging the isoFLOP cost calculation provided in  [3], we can estimate the inference cost of our models using this formula: 2*N*D+2*N*sqrt(D)*L
>
>  N: number of parameters, D: number of input tokens, L: number of decoding steps
>
> From our rejection rate, we can compute the expected rejection rate of gemma-2b-it during the training process.  We use 10% as our expected rejection rate (maybe slightly higher than actual rejection rate during training but easier for analysis). Therefore, the expected acceptance rate is 90%. We can estimate expected generated tokens under one run of line 5. Algorithm 1 shows a simplified algorithm where gamma=1. In practice, we use gamma=5.
>
> We can use geometric series to estimate expected generated tokens similar to [4]. E(# generated tokens with one run)=(1-0.9^5)/(1-0.9)=4.1. For simplicity of analysis, we round it to 4. For 5 student proposed tokens, the teacher evaluates all tokens in parallel  and likely rejects the last one. Therefore, we can approximately claim that in SKD, the student model will generate 80% of tokens while the teacher will generate 20% of them. To put this into inference time formula,
>
> Assuming N_student=2B, N_teacher=7B, D=256, L=200, ground truth is not available and needs to be sampled from the teacher (practical case).
>
> Cost (on-policy) = 2*2B*256+2*2B*16*200=13,824B
>
> Cost (SKD) = 2*2B*256+2*2B*16*200 * 80%+ 2 * 7B * 256 + 2*7B*16*200 * 20% = 23,808B
>
> Cost (Supervised KD) = 2 * 7B * 256 + 2*7B*16*200 = 48,384B
>
> Sampling cost of SKD is roughly 1.72 x bigger than on-policy KD, supervised KD is roughly 3.5 x bigger than on-policy KD. **SKD can save roughly 50% of compute from directly sampling from the teacher.**
>
> Note that we approximately calculate this compute using an instruction-tuned student model. If we use the SFTed student model, more computes can be further saved. This is mostly to give the reviewer a rough overview of how much computes that we needed for SKD sampling.
>
> [3] Scaling Laws for Neural Language Models
>
> [4] Fast Inference from Transformers via Speculative Decoding

---

> ### Author Response · Authors · 2024-11-21
> **Response to Reviewer EjjZ regarding to weakness b)**
>
> Corresponds to "(b) Moreover, I think the value for K should be studied. Essentially for high Ks the method should get somewhat similar to On Policy KD and for lower ones should be closer to Supervised KD? So there should be a sweet spot. Also I’m wondering if the value for K needs to be adjusted based on the training step? In other words, maybe higher or lower Ks would be better at the early or later stages of distillation? Some ablation experiments (with different numbers for K) could be helpful in addressing this."
>
> Thanks to the reviewer for bringing up the ablation of the value of K. We studied the effect of different Ks across different tasks at Appendix B.  Our findings are threefold: 1) The overall performance trend confirms our hypothesis that as K increases, SKD degenerates to on-policy KD like behavior, while decreasing K degenerates SKD to supervised KD like behavior. Consequently, neither the highest nor the lowest values of K are optimal. 2) We can find a sweet spot for K across tasks (Appendix B). Although for generalization purposes, we didn’t over-optimize k for each task. 3) We also showed a wide range of Ks can outperform both supervised KD and on-policy KD.
>
> We also explored experiments by adjusting K during training steps (decreasing K). However, we didn’t obtain positive signals (it didn’t beat constant K approach). Our hypothesis is that SKD assumes that as the model improves, it naturally acts like on-policy KD. However, heuristically decreasing K can force it to replace the token and behave like supervised KD, which introduces off-policy tokens, hurting the performance of the model. However, we encourage future works to explore more into this adaptive K direction.
>
>
> Finally, we conducted experiments using adaptive K strategy at decoding steps. Due to the autoregressive nature of our base models, their token generation becomes more deterministic with longer sequences. This raises the question of whether reducing K as the sequence grows could further improve performance. We experimented with a linear decay of K, decreasing it by $1$ for each additional generated token. We set a minimum K to ensure our algorithm does not degenerate to supervised KD towards the end of decoding.
>
>
> |         |  Translation  |  GSM8K  |
> |----------------|--------|----------------|
> |   SKD(K=25)  |  75.3   |      29.1     |
> |     SKD(K from 50 to 25)      |  75.1  |      25.2      |
> |     SKD(K from 25 to 15)      |  75.0  |      28.7      |
>
> We didn’t receive strong positive signals through this approach. We believed that future works could explore more into this direction since this is still a naive adaptive k scheme.

---

> ### Author Response · Authors · 2024-11-21
> **Response to Reviewer EjjZ regarding to weakness from c-g**
>
> Thank you so much for your suggestion! We will revise each part accordingly!
>
> Regarding to "c) In Line 119, it is stated that “the temperature t introduces randomness”. I think this is not entirely correct as it’s the sampling that introduces randomness. The temperature adjusts the distribution so it adjusts the randomness (as it is also stated correctly in the following sentence)."
>
> We revised phrase in Line 119 and mentions that "The temperature adjusts the distribution. Therefore, it adjusts the randomness when we sample from the model".
>
> Regarding to "(d) In Line 184, “Correct previous mistakes”, it is not clear what mistake is referring to. Is a mistake a poorly generated Token? What does correcting refer to?"
>
> We revised our writing at Line 184. "previous mistakes" refers to "poor tokens" that generated  from student model
>
> Regarding to "(f) In the pseudo-code (Algorithm 1), some variables, such as alpha in line 3, are not defined."
>
> alpha is defined at line 164. We will make it more clear in the paper.
>
> Regarding to "(g) This is not really an issue, rather a suggestion. A tiny figure to explain concepts defined in Section 3.1 would be very helpful. Basically a 1-row figure divided into maybe 4-5 sub-columns, with every column showing how each of the prior work applies the loss at token-level. Maybe something like Figure 1 in https://arxiv.org/pdf/2104.09044 I think Figure (2) can be refactored a bit so that it also includes other prior work. "
>
> Thank you so much for this suggestion! We will revise accordingly!

---

> ### Author Response · Authors · 2024-11-21
> **Response to Questions**
>
> (a) It is not 100% clear to me why would one replace a bad token with a token sampled from teacher's distribution. Could the authors give some intuition of why not re-attempt the sampling from student's distribution (line 4 in Algorithm 1) to replace the last token but under the condition that it's within TopK of the teacher? In otherwords, maybe the bad token sampled from student's distribution can be replaced with a better token that is sampled from the same distribution, rather than one that comes from the teacher's. If it is within the computational budget, an experiment to demonstrate this could also help prove the point.
>
> This is an implementation choice. For example, top-K tokens among teacher vocabulary can have a cumulative probability of 0.1 at student’s vocab distribution. Yes, we can continuously resample from the student model until we yield a token within the teacher's top-K. It still will be an off-policy token because it is not the top token likely to be generated by students.
>
> (b) If I understood correctly, your method does not add any computational overhead to the distillation process, as it is a simple probability check and a token replacement. If so, please assert this both in rebuttal and also in the paper. Otherwise, a clear discussion on computational overhead is missing.
>
> Thanks! Please refer to the response to weakness a). We will include these details in our final version.

---

> ### Author Response · Authors · 2024-11-23
> **Additional questions?**
>
> Hi Reviewer EjjZ,
>
> Thank you again for your time and efforts in our work and your feedback is quite constructive to revise our paper! We have meticulously considered and responded to each of your concerns. If you have any additional questions, we are very happy to address them during this open discussion period.
>
> Best regards,
>
> The authors

---

> ### Author Response · Authors · 2024-11-23
> **Possibility of ajusting the confidence score**
>
> We kindly request if the reviewer could consider raising the overall rating or confidence score, provided there are no additional concerns.
>
> Best,
> Authors

---

### Official Review · Reviewer_cyWE · 2024-11-03

**Soundness:** 3
**Presentation:** 4
**Contribution:** 2
**Rating:** 6
**Confidence:** 4

**Summary:**

The authors present Speculative Knowledge Distillation (SKD), a new KD method for effective knowledge transfer from large teacher models to smaller student models. SKD combines on-policy sampling with speculative decoding, allowing students to propose tokens while the teacher verifies and refines them, addressing issues in traditional KD methods like distribution mismatches and low-quality samples.

The authors evaluate SKD across tasks including translation, summarization, math, and instruction following, consistently outperforming baseline methods across different domains, data sizes, and initialization strategies. SKD shows robust performance in both task-specific and task-agnostic distillation, with strong generalization on complex math tasks and advantages in low-data scenarios.

Contributions:
1. Introducing SKD, which integrates on-policy sampling and speculative decoding for adaptive, high-quality knowledge transfer.
2. Demonstrating SKD’s superior performance across varied tasks, outperforming supervised and on-policy KD.
3. Validating SKD’s adaptability and efficiency, especially in diverse model initializations and low-data settings, without requiring additional supervised fine-tuning.

**Strengths:**

1. The paper introduces Speculative Knowledge Distillation (SKD) to address the distribution mismatch between training and inference in traditional knowledge distillation. By combining on-policy sampling with speculative decoding, SKD enables efficient knowledge transfer, allowing the student model to autonomously generate and refine its samples.

2. The authors validate SKD across various task scenarios, including translation, summarization, arithmetic reasoning, and instruction following, covering both task-specific and task-agnostic applications. Through comparisons with multiple baseline methods (e.g., supervised KD, on-policy KD, and ImitKD), the experimental setup is comprehensive and broadly applicable, providing certain support for SKD’s performance.

3. SKD’s design builds on imitation learning theory, lending theoretical validity to the approach. Additionally, the paper includes detailed implementation steps and hyperparameter settings, providing actionable guidance for future research.

**Weaknesses:**

1. Impact of Early Low-Quality Samples Lacks In-Depth Analysis: The SKD framework uses interleaved sampling and speculative decoding to handle low-quality samples generated by the student model during its initial training stages. However, the authors do not analyze how these low-quality samples may impact complex tasks, such as mathematical reasoning. Early low-quality samples could affect the stability of teacher feedback, slow convergence, and compromise overall stability. The authors could consider adding an analysis of model convergence behavior across tasks of varying complexity in their experiments to gain a more comprehensive understanding of SKD’s convergence characteristics in complex scenarios.

2. Task-Specific Adaptability: SKD maintains overall sampling quality through dynamic adjustments of token-level sample quality and adaptive switching between teacher and student sampling. However, the authors do not explore how to customize these mechanisms for different task types or complexities.

3. Effect of Model Initialization Lacks Discussion: While the authors examine SKD’s performance under different initialization conditions, such as instruction tuning and supervised fine-tuning, they do not discuss how initial model quality affects SKD’s convergence efficiency and stability. The authors might consider investigating convergence behavior under various initialization qualities, particularly in low-quality scenarios, by dynamically adjusting the proportion of teacher samples to stabilize the training process.

**Questions:**

Question 1: The paper does not analyze how early low-quality samples generated by the student model may impact complex tasks, such as mathematical reasoning. How do the authors believe these early samples could affect the stability of teacher feedback and convergence rates?

Question 2: Although the authors' framework incorporates dynamic adjustments and adaptive switching between teacher and student sampling, how do they envision customizing these mechanisms for different task types or complexities?

Question 3: While the authors examine SKD’s performance under various initialization conditions, how does initial model quality affect convergence efficiency and stability?

---

> ### Author Response · Authors · 2024-11-21
> **Response to Reviewer cyWE regarding to weakness 1)**
>
> Responding to "Impact of Early Low-Quality Samples Lacks In-Depth Analysis: The SKD framework uses interleaved sampling and speculative decoding to handle low-quality samples generated by the student model during its initial training stages. However, the authors do not analyze how these low-quality samples may impact complex tasks, such as mathematical reasoning. Early low-quality samples could affect the stability of teacher feedback, slow convergence, and compromise overall stability. The authors could consider adding an analysis of model convergence behavior across tasks of varying complexity in their experiments to gain a more comprehensive understanding of SKD’s convergence characteristics in complex scenarios."
>
> Thank you so much for your constructive feedback. We included analysis below.
>
> Table below shows validation loss respect to training steps at summarization (**For IT initialization**)
>
> | Steps          |  SKD  |  On-policy  |  Supervised KD  |
> |----------------|--------|----------------|----------------|
> |   0  |  4.44  |      4.44      |      4.44      |
> |   50  |  0.54  |     0.60      |      0.55     |
> |   100  |  0.46  |     0.50      |      0.45     |
> |   200  |  0.42  |     0.44      |      0.42     |
> |   300  |  0.40  |     0.42      |      0.42     |
> |   350  |  0.40  |     0.42      |      0.42     |
>
> Table below shows validation loss respect to training steps at GSM (**For IT initialization**)
>
> | Steps          |  SKD  |  On-policy  |  Supervised KD  |
> |----------------|--------|----------------|----------------|
> |   0  |  2.32  |      2.32      |      2.32      |
> |   50  |  0.27  |     0.44      |      0.39     |
> |   100  |  0.23  |     0.28      |      0.32     |
> |   200  |  0.21  |     0.25      |      0.28     |
> |   300  |  0.20  |     0.23      |      0.27     |
> |   350  |  0.19  |     0.22      |      0.26     |
>
> Table below shows validation loss respect to training steps at Translation (**For IT initialization**)
>
> | Steps          |  SKD  |  On-policy  |  Supervised KD  |
> |----------------|--------|----------------|----------------|
> |   0  |  5.96  |      5.96      |      5.96      |
> |   50  |  1.35  |     2.32      |      1.16     |
> |   100  |  1.06  |     2.09      |      0.99     |
> |   200  |  0.91  |     2.05      |      0.95    |
> |   300  |  0.87  |     2.00      |      1.04     |
> |   350  |  0.83  |     1.91      |      1.03     |
>
> We report the validation loss of various baseline KD strategies across 350 training steps, distilling Gemma-7B to Gemma-2B-it model. Our findings demonstrate that SKD consistently achieves the lowest validation loss by the final iteration across all three tasks. We note an important observation regarding on-policy samples: their lower quality can cause slow initial convergence (as seen at step 50 across all tasks), where on-policy KD exhibits a higher validation loss compared to both supervised KD and SKD. In translation tasks, the student models trained with on-policy KD continue to show higher validation losses than those trained with SKD or supervised KD due to the inferior quality of on-policy samples, as reflected by the lower COMET scores detailed in Table 1. For simpler tasks, like translation and summarization, supervised KD can achieve lower initial validation losses compared to both SKD and on-policy KD. However, SKD rapidly improves in later iterations, eventually reaches the lower validation loss compared to two baselines by the end of the training process, where supervised KD has the risk of overfitting (shown at translation table). Conversely, for more complex tasks, such as the GSM task, SKD’s interleaved sampling method enables rapid convergence and consistently lower validation losses throughout the training process, outperforming both on-policy and supervised KD.

---

> > ### Comment · Reviewer_cyWE · 2024-11-24
> > **Score Changed**
> >
> > Regarding weaknesses 1 and 3:
> > The authors' validation loss analysis across different tasks (summarization, GSM, and translation) and initialization conditions (IT and SFT) provides strong evidence of SKD's effectiveness. I suggest adding these results to the supplement and referencing them in the main paper.
> >
> > Regarding weakness 2:
> > This was my oversight, as the authors have provided a detailed analysis in Appendix B, demonstrating how K values correlate with task characteristics. I appreciate their investigation of this aspect.
> >
> > I thank the authors for their detailed reply, which has addressed my concerns. Accordingly, I have increased my score to 6.

---

> > > ### Author Response · Authors · 2024-11-26
> > > **Thanks for the reply**
> > >
> > > Thanks a lot for your suggestion and score updates! We will update paper accordingly.

---

> ### Author Response · Authors · 2024-11-21
> **Response to Reviewer cyWE regarding to weakness 2)**
>
> Responding to "Task-Specific Adaptability: SKD maintains overall sampling quality through dynamic adjustments of token-level sample quality and adaptive switching between teacher and student sampling. However, the authors do not explore how to customize these mechanisms for different task types or complexities."
>
> Thank you to the reviewer for highlighting the importance of the ablation study on the value of K. We have detailed the effects of varying K across different tasks in Appendix B of our paper. Our findings are threefold: 1) The overall performance trend confirms our hypothesis that as K increases, SKD degenerates to on-policy KD like behavior, while decreasing K degenerates SKD to supervised KD like behavior. Consequently, neither the highest nor the lowest values of K are optimal. 2) We can find a sweet spot for K across tasks (Appendix B, K=50 for translation, K=5 for summarization and K=25 for GSM at Figure 6). For generalization purposes, we didn’t over-optimize k for each task. 3) We also showed a wide range of Ks can outperform both supervised KD and on-policy KD.
>
> We suspect that a higher K value may be optimal for more constrained, close-ended tasks such as translation. This setting allows the student model to explore a range of sample qualities, both high and low, to ensure better coverage and diversity within a limited sample space. For open-ended tasks such as conversation summarization, we recommend a lower K value in SKD. Higher K values may cause students to explore an excessive range of answer options,  potentially diverting the student model’s learning trajectory. By reducing K, we facilitate more frequent interleaving of teacher sampling, which helps ensure that the samples maintain meaningful learning signals. We leave more rigorous study to future research.

---

> ### Author Response · Authors · 2024-11-21
> **Response to Reviewer cyWE regarding to weakness 3)**
>
> Corresponding to "Effect of Model Initialization Lacks Discussion: While the authors examine SKD’s performance under different initialization conditions, such as instruction tuning and supervised fine-tuning, they do not discuss how initial model quality affects SKD’s convergence efficiency and stability. The authors might consider investigating convergence behavior under various initialization qualities, particularly in low-quality scenarios, by dynamically adjusting the proportion of teacher samples to stabilize the training process."
>
> Table below shows validation loss respect to training steps at Summarization (**For SFTed initialization**)
>
> | Steps          |  SKD  |  On-policy  |  Supervised KD  |
> |----------------|--------|----------------|----------------|
> |   0  |  1.56 |      1.56      |     1.56      |
> |   50  |  0.52  |     0.54      |      0.51     |
> |   100  |  0.44  |     0.47      |      0.44     |
> |   200  |  0.41  |     0.43      |      0.42     |
> |   300  |  0.40  |     0.42      |      0.42     |
> |   350  |  0.40  |     0.42      |      0.42     |
>
> Table below shows validation loss respect to training steps at GSM (**For SFTed initialization**)
>
> | Steps          |  SKD  |  On-policy  |  Supervised KD  |
> |----------------|--------|----------------|----------------|
> |   0  |  0.244  |      0.244      |     0.244      |
> |   50  |  0.214  |     0.215      |      0.208     |
> |   100  |  0.202  |     0.204      |      0.200     |
> |   200  |  0.188  |     0.195      |      0.191     |
> |   300  |  0.182  |     0.183      |      0.189     |
> |   350  |  0.178  |     0.180      |      0.188     |
>
> Table below shows validation loss respect to training steps at Translation (**For SFTed initialization**)
>
> | Steps          |  SKD  |  On-policy  |  Supervised KD  |
> |----------------|--------|----------------|----------------|
> |   0  |  1.14  |      1.14      |     1.14      |
> |   50  |  0.886  |     0.891      |      0.997     |
> |   100  |  0.859  |     0.862      |      0.948     |
> |   200  |  0.843  |     0.832      |      0.980     |
> |   300  |  0.812  |     0.831      |      0.929     |
> |   350  |  0.807  |     0.813      |      0.908     |
>
> We report the validation loss of various baseline KD strategies across 350 training steps, distilling Gemma-7B to SFTed Gemma-2B model.  Our findings demonstrate that SKD consistently achieves the lowest validation loss by the final iteration across all three tasks. Our finding shows that as base quality improves, on-policy KD consistently achieves lower validation loss compared to supervised KD at the end of training steps. This suggests that on-policy approaches can achieve better convergence when sample quality is improved. Most importantly, under both model initialization, SKD can achieve the best convergence with 350 training steps compared to both supervised KD and on-policy KD.
>
> Our training process remains stable across on-policy KD, supervised KD, and general KD because KL loss provides a relatively stable training objective. The key distinction between these approaches lies in the effectiveness of the samples for learning, as measured by their ability to minimize validation loss. Through comparative analysis of validation losses from both instruction-tuned (IT) and supervised fine-tuned (SFT) student model initializations, we have demonstrated that SKD achieves superior validation convergence compared to the baseline methods, by achieving the lowest validation loss.

---

> ### Author Response · Authors · 2024-11-23
> **Additional questions?**
>
> Hi Reviewer cyWE,
>
> Thank you again for your time and efforts in our work and your feedback is quite constructive to revise our paper! We have meticulously considered and responded to each of your concerns. If you have any additional questions, we are very happy to address them during this open discussion period. If you don't have additional concerns, would you kindly consider raising the score? Thanks again for your time and efforts!
>
> Best regards,
>
> The authors

---

### Official Review · Reviewer_VY1t · 2024-11-04

**Soundness:** 3
**Presentation:** 4
**Contribution:** 3
**Rating:** 6
**Confidence:** 5

**Summary:**

This paper proposed a new knowledge distillation method called Speculative KD (SKD), inspired by speculated decoding, that addresses the drawbacks from supervised KD and on-policy, while keeping the advantages of both worlds. Specifically, to circumvent the issue of supervised KD, SKD enforces the teacher model to evaluate the output of the student model. To alleviate the "cold-start" OOD problem of on-policy KD, SKD filters out intermediate tokens that the teacher is unlikely to generate, and re-samples from the teacher.

**Strengths:**

- The motivation of the paper is convincing. Supervised KD is known to suffer from distribution shift, while on-polich KD suffers from the OOD problem.
- The organization is well and the paper is carefully written. The problem setting is well formulated.
- All-around empirical results pretty much validates the effectiveness of the method. The authors conducted experiments on four tasks and one task agnostic distillation.

**Weaknesses:**

- Although it is relatively straightforward to agree with the authors that the on-policy KD suffer from the OOD issue, is there a way to show how severe the issue is? For example, would it be possible to quantitatively evaluate?
- It is a bit hard to understand the task-agnostic distillation setting. I see the training and test sets are both based on the math reasoning datasets. Where does the shifts come from?
- In table 2, it is obvious that the improvements are rather marginal. In Table 1, it seems the proposed method falls short behind another baseline. Would you explain?
- Significant tests for Table 1 & 2 could further strengthen the results.

**Questions:**

- Is there a way to show how severe the OOD issue of on-policy KD is? For example, would it be possible to quantitatively evaluate?
- Why does the setting proposed in Section 5.2 a "task-agnostic" one?

---

> ### Author Response · Authors · 2024-11-21
> **Response to Reviewer VY1t regarding to weakness 1)**
>
> Correspond to "Although it is relatively straightforward to agree with the authors that the on-policy KD suffers from the OOD issue, is there a way to show how severe the issue is? For example, would it be possible to quantitatively evaluate?"
>
> The following table is Table1
> | From SFTed Gemma-7B  |  On-policy (student only)  |  SKD (teacher and student interleaved)  |  Supervised KD (ground truth)  |
> |----------------|--------|----------------|----------------|
> |  Teacher PPL |  46.8   |      10.8     |      1.57      |
>
>
> Thanks for the insightful feedback. We followed the suggestion and calculated the per-sample perplexity (PPL) using both the teacher (supervised FT Gemma-7B) and the student (instruction-tuned Gemma-2B). Our analysis included ground truth data, on-policy samples proposed by the student, and samples generated by SKD. We randomly selected100 data points from translation training data for student and SKD sampling. For supervised KD, we used ground truth output.
>
> As shown in Table1, the teacher model assigned a significantly higher PPL (46.8 vs 1.57) to student-proposed samples compared to ground truth, indicating a substantial divergence from the teacher's distribution. Notably, SKD-proposed samples achieved a much lower PPL (10.8), suggesting that the teacher model can more accurately assess the quality of SKD-generated samples.
>
> The following table is Table2
>
> | From Gemma-2B-it  |  On-policy (student only)  |  SKD (teacher and student interleaved)  |  Supervised KD (ground truth)  |
> |----------------|--------|----------------|----------------|
> |  Student PPL |  44.0   |      358     |      5606.3      |
>
> Table 2 demonstrates that ground truth samples exhibit high PPL (5606.3) under the student model, confirming their off-policy nature. Crucially, our SKD approach, which combines on-policy token proposals with teacher corrections, substantially reduces PPL (358) compared to supervised KD. This indicates that SKD samples are more aligned with the student model's distribution, facilitating effective learning.

---

> ### Author Response · Authors · 2024-11-21
> **Response to Reviewer VY1t regarding to weakness 2)**
>
> Corresponding to "It is a bit hard to understand the task-agnostic distillation setting. I see the training and test sets are both based on the math reasoning datasets. Where does the shifts come from?"
>
> In task-agnostic distillation, the model is trained on a diverse set of tasks within a domain (in this case, mathematics) to develop general capabilities. However, it is then evaluated on specific tasks within that domain that were not seen during training. In our study, the MathInstruct model was trained on a variety of math datasets (MathQA [2], math reasoning [3], tabular processing [4]) to learn general mathematical skills. We then tested its ability to generalize to new, unseen math problems in the ASDIV [5], SVAMP [6], and GSM_plus [7] datasets. This approach, where the model is trained on a broad set of tasks and tested on specific ones, is consistent with the task-agnostic distillation framework described in [1].
>
> [1] Advancing LLM Reasoning Generalists with Preference Trees
>
> [2] MathQA: Towards interpretable math word problem solving with operation-based formalisms
>
> [3] NumGLUE: A suite of fundamental yet challenging mathematical reasoning tasks.
>
> [4] Dynamic prompt learning via policy gradient for semi-structured mathematical reasoning
>
> [5] A diverse corpus for evaluating and developing English math word problem solvers
>
> [6] Are NLP models really able to solve simple math word problems?
>
> [7] Gsm-plus: A comprehensive benchmark for evaluating the robustness of llms as mathematical problem solver

---

> ### Author Response · Authors · 2024-11-21
> **Response to Reviewer VY1t regarding to weakness 3) and 4)**
>
> Corresponding to "In table 2, it is obvious that the improvements are rather marginal. In Table 1, it seems the proposed method falls short behind another baseline. Would you explain? Significant tests for Table 1 & 2 could further strengthen the results."
>
> Thanks for the question. We conducted a permutation significance test for SKD against all baselines for translation and summarization. SKD outperforms all baselines (p<0.05) except for SKD vs ImitKD at Gemma-7B to Gemma-2B-IT on summarization data and SKD vs supervised KD/SFT at Qwen-7B to Qwen-0.5B-it on translation data.
>
> We would like to clarify the scale of our metrics used in the evaluation (See Appendix F). COMET-22 is a learned metric from human rating data. Despite its high correlation to humans, its score value is not directly interpretable as in traditional accuracy metrics. From Table2, SKD improves the best baseline at SFTed Gemma-2b and SFTed Qwen2-0.5B, with score difference 0.4 and 0.7 respectively. Referring to the recent paper [2], under 0.4 score difference, 75% metric decisions will be aligned with human judgments, and under 0.7 score difference, 85% metric decisions will be aligned with human judgments. This indicates SKD's improvements are significant.
>
> Regarding your comment about Table 1, where you mentioned that “it seems the proposed method falls short behind another baseline,” We believe the reviewer is referring to our results from Qwen7B to Qwen-0.5B-it translation. Table 1 shows that Qwen-0.5B-it’s performance under Assamese-to-English translation is extremely poor. Therefore, with this poor model initialization, on-policy or imitKD can have deteriorated performance due to low quality student samples. On the other hand, samples coming from supervised KD/SFT have advantages due to much better quality samples.  We show that SKD can dynamically adjust sampled tokens to achieve close performance to supervised KD/SFT. In practice, we will use SFTed initialized Qwen-0.5B model as the student. From Table 2, we showed that SKD outperforms all baselines and beats the best baseline ImitKD by 0.7 COMET score.
>
> [2] Navigating the Metrics Maze: Reconciling Score Magnitudes and Accuracies

---

> ### Author Response · Authors · 2024-11-23
> **Additional questions?**
>
> Hi Reviewer VY1t,
>
> Thank you again for your time and efforts in our work and your feedback is quite constructive to revise our paper! We have meticulously considered and responded to each of your concerns. If you have any additional questions, we are very happy to address them during this open discussion period.
>
> We hope that most of the reviewer’s concerns have been addressed and, if so, they would reconsider their assessment. We’d be happy to engage in further discussions.
>
> Best regards,
> The authors

---

> ### Author Response · Authors · 2024-11-25
> **Additional concern if any**
>
> Hi Reviewer VY1t,
>
> Thank you again for your time and efforts in our work and your feedback is quite constructive to revise our paper! We have addressed all reviewer cyWE and  EjjZ's concerns (cyWE raised score for us). As discussion period moves towards end, is there any additional concern that you would like us to address? We are happy to engage in further discussion.
>
> Best,
>
> Authors

---

> > ### Comment · Area_Chair_ehY4 · 2024-12-02
> > **Did you review the responses? Any concerns please?**
> >
> > Dear Reviewer
> > As we are getting closer to the review period, please engage in the discussion with the authors and let us know if you have further questions or are the responses satisfactory?
> >
> > Thanks!

---

> ### Comment · Reviewer_VY1t · 2024-12-03
> **Reviewer Response**
>
> Dear Authors,
>
> Thank you for the further clarification, especially the evaluation of quantifying the OOD issue suffered by on-policy KD. Considering the technical contribution and soundness, I think score 6 is appropriate. I have raised my confidence score to reflect my evaluation.

---

### Author Response · Authors · 2024-12-03
**Discussion phase summary**

We again thank all reviewers for their helpful reviews and comments. We are glad to see all reviewers give us positive feedback after discussion period, showing reviewers' genuine interest in our work.

During the rebuttal period, we addressed all questions from all three reviewers. Each reviewer acknowledged our responses and expressed no further concerns.

To conclude the discussion phase, we briefly  summarize the main concerns raised in the reviews and corresponding improvements in the paper.

**Quantify OOD issues for poor student samples**

 To address concerns from reviewer VY1t,  we utilized teacher perplexity (PPL) to quantify out-of-distribution (OOD) issues in poor on-policy samples. We clearly demonstrate OOD issues using PPL.

**Show SKD's convergence compared to on-policy and supervised KD**

To address concerns from reviewer cyWE,  we demonstrated the effectiveness of SKD by showing that it achieves superior validation convergence compared to both on-policy KD and supervised KD under instruction tuned and SFTed initialization across three tasks.

**Show student token rejection rate during SKD training process and estimate SKD's computation**

To address concerns from EjjZ, we charted the teacher model’s rejection rates against tokens proposed by the student model during the training process and calculated the theoretical computational demands of SKD compared to sampling from student and teacher models. SKD can save roughly 50% of compute from directly sampling from the teacher.

**Ablation study of K**

We also pinpoint reviewer cyWE and EjjZ for the ablation of K results in Appendix B. In the section, we showed a wide range of Ks can outperform both supervised KD and on-policy KD. We can find the most optimized K across tasks. However, for generalization purposes, we didn’t over-optimize k for each task.


**Additional revisions**

Aside from main discussion listed above, we also addressed some smaller comments and suggestions from the reviewers.

We will incorporate these updates into the camera-ready version of the paper.

Overall, we believed that reviewer's comments helped us greatly improved paper quality. We thank reviewers for your efforts during rebuttal and discussion period.

Best,

Authors

---

### Meta-Review · Area_Chair_ehY4 · 2024-12-18

**Metareview:**

The paper presents a Knowledge Distillation method that circumvents issues in both supervised KD and on-policy KD [Reviewer VY1t]

The authors evaluate SKD across several types of tasks including translation, summarization, math, and instruction following, consistently outperforming baseline methods across different domains, data sizes, and initialization strategies. SKD shows robust performance in both task-specific and task-agnostic distillation, with strong generalization on complex math tasks and advantages in low-data scenarios. [Reviewer cyWE]

The idea is clearly motivated in the paper and makes sense. The quantitative results also seem promising (across tasks the method is mostly outperforming prior work by a good margin). Especially the method is consistently outperfoming On Policy KD, which supports the proposed idea of switching to teacher tokens if the student tokens are bad. [Reviewer EjjZ]

Authors have addressed concerns from reviewers and have updated the manuscript with sufficient empirical results to substantiate claims and to enhance the robustness, scalability and convergence of the proposed SKD.

**Additional Comments On Reviewer Discussion:**

There have been intense discussions during the rebuttal phase and the authors rebuttal has answered most of the concerns raised by the reviewers. Authors have also improved the manuscript and the supplementary materials to improve the quality of the paper - to include discussion on computational overhead (the computation of this overhead is not straightforward as agreed by Reviewer EjjZ), Quantified OOD issues for poor student samples (reviewer VY1t), Showed SKD's convergence compared to on-policy and supervised KD (reviewer cyWE), included ablation studies (Reviewers cyWE and EjjZ for the ablation of K results)

All the three reviewers have increased their score and/or confidence after the rebuttal phase. This paper can clearly be accepted as a poster.

---

### Decision · Program_Chairs · 2025-01-22

Accept (Poster)